# Assessment of antimalarial activity of crude extract of Chan-Ta-Lee-La and Pra-Sa-Chan-Dang formulations and their plant ingredients for new drug candidates of malaria treatment: *In vitro* and *in vivo* experiments

**Prapaporn Chaniad[1,2], Arisara Phuwajaroanpong[1,2], Walaiporn Plirat[1,2], Atthaphon Konyanee[1,2], Abdi Wira Septama[3], Chuchard Punsawad [1,2]***

1 Department of Medical Sciences, School of Medicine, Walailak University, Nakhon Si Thammarat, Thailand, 2 Research Center in Tropical Pathobiology, Walailak University, Nakhon Si Thammarat, Thailand, 3 Research Center for Pharmaceutical Ingredient and Traditional Medicine, Cibinong Science Center, National Research and Innovation Agency (BRIN), West Java, Indonesia

* chuchard.pu@wu.ac.th

## Abstract

The emergence and spread of antimalarial drug resistance have become a significant problem worldwide. The search for natural products to develop novel antimalarial drugs is challenging. Therefore, this study aimed to assess the antimalarial and toxicological effects of Chan-Ta-Lee-La (CTLL) and Pra-Sa-Chan-Dang (PSCD) formulations and their plant ingredients. The crude extracts of CTLL and PSCD formulations and their plant ingredients were evaluated for *in vitro* antimalarial activity using *Plasmodium* lactate dehydrogenase enzyme and toxicity to Vero and HepG2 cells using the tetrazolium salt method. An extract from the CTLL and PSCD formulations exhibiting the highest selectivity index value was selected for further investigation using Peter's 4-day suppressive test, curative test, prophylactic test, and acute oral toxicity in mice. The phytochemical constituents were characterized using gas chromatography-mass spectrometry (GC-MS). Results showed that ethanolic extracts of CTLL and PSCD formulations possessed high antimalarial activity (half maximal inhibitory concentration = 4.88, and 4.19 g/mL, respectively) with low cytotoxicity. Ethanolic extracts of the CTLL and PSCD formulations demonstrated a significant dose-dependent decrease in parasitemia in mice. The ethanolic CTLL extract showed the greatest suppressive effect after 4 days of suppressive (89.80%) and curative (35.94%) testing at a dose of 600 mg/kg. Moreover, ethanolic PSCD extract showed the highest suppressive effect in the prophylactic test (65.82%) at a dose of 600 mg/kg. There was no acute toxicity in mice treated with ethanolic CTLL and PSCD extracts at 2,000 mg/kg bodyweight. GC-MS analysis revealed that the most abundant compounds in the ethanolic CTLL extract were linderol, isoborneol, eudesmol, linoleic acid, and oleic acid, whereas ethyl 4-methoxycinnamate was the most commonly found compound in the ethanolic PSCD extract, followed by 3-hydroxy-2-(4-hydroxy-3-methoxyphenyl)-4H-chromen-4-one, flamenol, oleic acid amide, linoleic acid, and oleic acid. In conclusions, ethanolic CTLL and PSCD extracts exhibited high

**Data Availability Statement:** The data used to support the findings of this study are included within the article.

**Funding:** This research was supported by the Agricultural Research Development Agency (Public Organization) (Contract No. PRP6505030080). The funders had no role in the study design, data collection and analysis, decision to publish, or manuscript preparation.

**Competing interests:** The authors have declared that no competing interests exist.

**Abbreviations:** CTLL, Chan-Ta-Lee-La; PSCD, Pra-Sa-Chan-Dang; pLDH, *Plasmodium* lactate dehydrogenase; MTT, tetrazolium salt; GC-MS, gas chromatography-mass spectrometry; $IC_{50}$, half maximal inhibitory concentration; ACTs, artemisinin-based combination therapies; RBCs, red blood cells; $CO_2$, carbon dioxide; PBS, phosphate-buffered saline; DMSO, dimethyl sulfoxide; ICR, Institute of Cancer; OECD, Economic Cooperation and Development; AST, aspartate aminotransferase; ALT, alanine aminotransferase; ALP, alkaline phosphatase; BUN, blood urea nitrogen; SEM, standard error of the mean; $CC_{50}$, half-maximal cytotoxicity concentration.

antimalarial efficacy *in vitro*. The ethanolic CTLL extract at a dose of 600 mg/kg exhibited the highest antimalarial activity in the 4-day suppressive and curative tests, whereas the ethanolic PSCD extract at a dose of 600 mg/kg showed the highest antimalarial activity in the prophylactic test.

## Introduction

Natural products play a significant role in drug discovery and the treatment of many diseases. Many natural products with diverse structures have various biological activities owing to their active ingredients, which can be used as therapeutic agents. Local communities have used natural products, especially medicinal plants, as traditional treatments for controlling multiple disorders, owing to their fewer adverse effects, affordability, and easy accessibility [1]. Herbal plant treatments involve the use of single or multiple components. However, combining multiple herbs or herbal formulas provides promising potential therapeutic effects and reduces toxicity compared with a single herb [2, 3]. Nowadays, natural products are still under consideration by the scientific community in the search for new therapies since the emergence of drug resistance is considered an urgent issue [4–6]. Failure to control or eliminate drug-resistant pathogens can also increase the spread of malaria. Malaria remains one of the most severe public health problems in tropical and subtropical regions. In 2022, malaria caused 0.6 million deaths worldwide, and approximately 241 million cases were reported in the World Malaria Report [7]. Artemisinin-based combination therapies (ACTs) are currently used as the first- and second-line treatments for uncomplicated *Plasmodium falciparum* malaria. Unfortunately, current available antimalarial drugs, such as artemisinin, chloroquine, sulfadoxine-pyrimethamine, atovaquone, proguanil, quinine, and doxycycline, have been reported to have reduced efficacy due to resistance to parasitic strains [8–11]. Delayed parasite clearance resulting from mutations is related to antimalarial drug resistance, leading to treatment failure [12]. Drug resistance, especially multidrug resistance, and treatment failure are the greatest challenges in controlling and eliminating malaria. Therefore, the demand for new antimalarial drugs to address this issue is extremely high. Plant-derived antimalarial agents, especially for herbal formulas, are an interesting approach because they have diverse structural and chemical compositions that act on multiple targets. A previous study suggested that the positive effects of crude plant extracts or a mixture of plants have higher in vitro and in vivo antiplasmodial activity than a single compound at an equivalent dose. Moreover, the combination strategy can reduce the toxicity of single-active compounds to the microorganism [13]. Therefore, this study is interested in examining the antimalarial activity of Chan-Ta-Lee-La (CTLL) and Pra-Sa-Chan-Dang (PSCD) formulations and their plant ingredients. CTLL consists of eight kinds of herbal plants, including *Atractylodes lancea* (Thunb.) DC., *Angelica sylvestris* L., *Artemisia vulgaris* L., *Dracaena loureiroi* Gagnep., *Eurycoma longifolia* Jack, *Gymnopetalum chinense* (Lour.) Merr., *Myristica fragrans* Houtt., and *Tinospora crispa* (L.) Hook.f. & Thomson [14]. PSCD consists of twelve herbs, including *Bouea macrophylla* Griff., *Caesalpinia sappan* L., *Citrus aurantiifolia* (Christm.) Swingle., *Dracaena cochinchinensis* (Lour.) S.C. Chen, *Heliciopsis terminalis* (Kurz) Sleumer, *Jasminum sambac* (L.) Aiton, *Kaempferia galanga* L., *Ligusticum chuanxiong* Hort., *Mammea siamensis* T. Anderson., *Mesua ferrea* L., *Myristica fragrans* Houtt., and *Nelumbo nucifera* Gaertn [15]. These formulations are well known for relieving fever in Thailand, and the formulations have been reported to possess antiulcerogenic, antibacterial, antiproliferative, and anti-inflammatory activities [14, 16, 17]. In addition, plant

ingredients made up for CTLL formulation have been reported that, *A. lancea* possess antimalarial activity against *Plasmodium berghei*-infected mice with a high parasite suppression of up to 60% [18]. *D. loureiroi* and *T. crispa* possessed a broad range of biological activities such as antioxidant, anti-inflammatory, antibacterial, antimalarial, antitumor, analgesic, hypoglycemic, and immune-enhancing effects [19, 20]. The biological activities of plant ingredients in PSCD formulation were reported that compounds isolated from *H. terminalis* exhibited anti-inflammatory, antioxidant, and hepatoprotective activities [21, 22]. *B. macrophylla* is scientifically reported to have antimicrobial, antihyperglycemic, antioxidant, anticancer, and antiphotoaging properties [23]. Limonoids and kaempferol, the important constituents of *C. aurantifolia* were reported to have anti-inflammatory, antimalarial, and protective activities for neurological disorders associated with cerebral malaria [24, 25]. *K. galangal* revealed a chemoprophylaxis effect against *P. berghei* infection in a mouse [26]. *M. fragrans* has confirmed that this plant contains a wide variety of pharmacological activities such as anticonvulsant, antioxidant, anti-inflammatory, hepatoprotective, antimicrobial, antiparasitic, and nematocidal activities, as well as a promising antimalarial activity both in vitro and in vivo [18, 27]. The chemical compounds from *C. sappan* and *Mammea siamensis* have been reported to have antimicrobial, anti-inflammatory, antimalarial, and neuroprotective properties [28, 29]. To the best of our knowledge, there has been no report on the antimalarial properties of CTLL and PSCD formulations. Furthermore, the various bioactive substances in CTLL and PSCD formulations may provide advantages for the treatment of malaria infections. The extracts may act directly on the parasite, promote an immune response to combat the parasite, or regulate the balance of the inflammatory response in this highly inflammatory disease. Therefore, this study aimed to screen the antimalarial activities and cytotoxic effects of plant ingredients, CTLL, and PSCD formulations using in vitro methods, and then confirm the antimalarial activities and toxic effects of the formulations in mouse models.

## Methods

### Ethics approval and consent to participate

Plant materials were collected in accordance with the relevant guidelines and regulations of the Plant Varieties Protection Department of Agriculture, Ministry of Agriculture and Cooperatives, Thailand. The study protocol was reviewed and approved by the Human Ethics Committee of Walailak University before the recruitment of participants (approval number: WUEC-22-099-01) and followed the tenets of the Declaration of Helsinki. Written informed consent was obtained from all participants before data and blood sample collection. The experimental protocols used in this study were reviewed and approved by the Animal Ethics Committee of Walailak University (WU-ACUC-65038). All research and animal care staff received appropriate training in animal care and the use of laboratory animals. All protocols in this study were performed in strict accordance with the relevant guidelines and regulations for using animals in compliance with the Animal Research: Reporting of In Vivo Experiments (ARRIVE) guidelines.

### Powder formulation and plant materials

The powder formulations of CTLL and PSCD were purchased from a drug store in Nakhon Pathom Province, Thailand. Eight plant ingredients for the CTLL formulation and 12 for the PSCD formulation were obtained from a traditional Thai medicine drug store in Nakhon Si Thammarat Province, Thailand (Table 1). Plant identification and authorization were evaluated based on morphological characteristics and confirmed by a botanist at the School of

**Table 1. List of traditional formulations used in this study.**

| Formulation | Plant ingredients | Family | Common name | Plant part | Voucher Number |
|---|---|---|---|---|---|
| CTLL | *Angelica sylvestris* L.. | Umbelliferae | Wild angelica | Roots | SMD276002003 |
| | *Atracylodes lancea* (Thunb) DC. | Asteraceae | Cang zhu | Rhizomes | SMD072010001 |
| | *Artemisia vulgaris* L. | Asteraceae | Mugwort | Whole plant | SMD024008001 |
| | *Myristica frgrans* Houtt. | Myristicaceae | Nutmeg | Stem | SMD177001003 |
| | *Dracaena loureiroi* Gagnep. | Agavaceae | Dragon blood | Stem | SMD096001007 |
| | *Gymnopetalum chinense* (Lour.) Merr. | Cucurbitaceae | | Fruit | SMD082007001 |
| | *Tinospora crispa* (L.) Hook.f. & Thomson | Menispermaceae | | Stem | SMD170016002 |
| | *Eurycoma longifolia* Jack. | Simarubaceae | Tongkat ali | Roots | SMD259003002 |
| PSCD | *Dracaena loureiri* Gagnep. | Dracaenaceae | Dragon blood | Stem | SMD096001007 |
| | *Heliciopsis terminalis* Sleum | Proteaceae | | Roots | SMD229003001 |
| | *Bouea macrophylla* Gritt. | Anacardiaceae | Marina plum | Roots | SMD007002001 |
| | *Citrus aurantiifolia* (Christm.) Swingle | Rutaceae | Lime | Roots | SMD252004001 |
| | *Kaempferia galanga* Linn. | Zingiberaceae | Cekur | Rhizomes | SMD288013007 |
| | *Ligusticum sinense* Oliv. | Apiaceae | Sichuan lovag | Rhizomes | SMD017003002 |
| | *Myristica fragrans* Houtt. | Myristicaceae | Nutmeg | Stem | SMD177001003 |
| | *Caesalpinia sappan* Linn. | Leguminosae | Sappan wood | Stem | SMD138007018 |
| | *Nelumbo nucifera* Gaertn. | Nelumbonaceae | Sacred lotus | Pollens | SMD181001001 |
| | *Mesua ferrea* Linn. | Guttiferae | Ceylon Ironwood | flower | SMD122007001 |
| | *Mammea siamensis* Kosterm. | Guttiferae | Negkassar | flower | SMD122006002 |
| | *Jasminum sambac* Ait. | Oleaceae | English Arabian jasmine | flower | SMD187007002 |

CTLL, Chan-Ta-Lee-La: CTLL; PSCD, Pra-Sa-Chan-Dang

Medicine of Walailak University. Voucher plant specimens have been deposited at the School of Medicine, Walailak University (Table 1).

## Plant preparation for crude extract

Plant extraction was performed as previously described [30]. First, the plant ingredients were washed with tap water and dried in a hot-air oven (Moment, Model; SFE600, Schwabach, Germany) for 72 h. Two traditional herbal formulations were prepared by mixing portions of the ingredients and weighing 60 g to obtain an adequate crude extract. Each plant and formulation were extracted using 95% ethanol and distilled water. An ethanolic solution was used as a solvent because it can dissolve several non-polar and polar molecules. Distilled water was selected as a solvent because it is commonly used in Thai folk medicine. For the ethanolic method, 60 g of each plant's sample or powder formulation was extracted in a flask containing approximately 600 mL of 80% ethanol at room temperature (25˚C) for 72 h. For the aqueous decoction method, the plant materials (60 g) or powder formulation were soaked in 600 mL of distilled water and boiled at 90–100˚ C for 30 min. Subsequently, the macerated solution from each protocol was permeabilized with gauze and Whatman No. 1 filter paper. The residues were re-extracted twice by adding an equal volume of fresh 80% ethanol or distilled water. The combined solutions were evaporated under reduced pressure using a rotary evaporator (Buchi® rotary evaporator, Model R-210, Shanghai, China) and lyophilized using a freeze dryer (Gamma 2–16 LSC-plus, Martin Christ, Osterode am Harz, Germany). The crude extracts were stored in a screwtop container and refrigerated at 4˚ C until use. Each crude extract was dissolved in 7% Tween 80 and 3% ethanol in distilled water or distilled water to provide a working solution for *in vitro* and animal experiments.

### *In vitro* culture of *Plasmodium falciparum*

The *P. falciparum* K1 strain was acquired from Dr. Rapatbhorn Patrapuvich of the Department of Drug Research Unit for Malaria at the Faculty of Tropical Medicine, Mahidol University in Thailand. The parasite was proliferated according to previously described methods with minor modifications [30, 31]. The parasite was grown in uninfected $O^+$ red blood cells (RBCs) with complete medium (RPMI-1640) containing cell culture supplements with 2 mg/mL sodium bicarbonate, 10 µg/mL hypoxanthine (Sigma-Aldrich, New Delhi, India), 4.8 mg/mL HEPES (HiMedia, Mumbai, India), 0.5% Albumax II (Gibco, Waltham, MA, USA), and 2.5 µg/mL gentamicin (Sigma-Aldrich). Culture flasks were incubated at 37° C and 5% carbon dioxide ($CO_2$) condition. The percentage of parasitemia was determined daily using a light microscope [32].

### Cell culture and maintenance

The HepG2 cell lines purchased from the American Type Culture Collection cell bank (HB-8065[TM]) and Vero cells (Elabscience, Wuhan, Hubei, China) were separately cultured in Dulbecco's Modified Eagle Medium with high glucose (Gibco, Carlsbad, CA, USA) supplemented with 10% fetal bovine serum (Sigma-Aldrich, New Delhi, India), 1% (v/v) penicillin/streptomycin (Sigma-Aldrich, St Louis, MO, USA), 1% HEPES (HyClone-Cytiva, Conroe, Texas, USA) at 37° C in an atmosphere of 5% $CO_2$. The morphology and confluence of both cell lines were examined daily using a phase-contrast inverted microscope (Olympus, Model CK X31, Hicksville, NY, USA). When the cells reached 80–90% confluence, 0.25% trypsin-ethylenediamine tetraacetic acid (Gibco, Carlsbad, CA, USA) was used to detach the cells for subculturing in a new flask.

### *In vitro* antimalarial activity testing

*In vitro Plasmodium* lactate dehydrogenase (pLDH) assay [33, 34] was used to test formulations and their plant ingredient extracts for antimalarial activity. The pLDH activity was determined by the concentration of crude extract that suppressed parasite proliferation by 50% [half maximal inhibitory concentration ($IC_{50}$)]. A stock solution (20 mg/mL) of the aqueous extract was dissolved in phosphate-buffered saline (PBS), whereas the ethanolic extract dissolved in dimethyl sulfoxide (DMSO). Two-fold serial dilutions of the crude extracts were performed using PBS for aqueous extracts or DMSO for ethanolic extracts, with a final concentration from 0.78 to 100 µg/mL. In this assay, 96 well plates containing 2% parasite culture were treated with crude extracts and incubated for 72 h at 37° C in a $CO_2$ incubator. Artesunate (Sigma-Aldrich, St Louis, MO USA) standard drug at concentration 0.78–100 µg/mL was added to each well as a positive control. PBS and DMSO (Merck, Darmstadt, Germany) were used as negative controls. Normal $O^+$ RBCs were used as blank controls. The experiment was performed in triplicate for each condition. After 72 h of incubation, the 96-well plate was frozen at -80° C and thawed at 37° C to lyse the RBCs. After that, 20 µL of suspension supernatant from each well was transferred to a new microplate to detect pLDH enzyme by the reaction between 100 µL of malstate reagent and 20 µL of nitroblue tetrazolium/phenazine ethosulfate solution (Calbiochem, Sigma-Aldrich, New Delhi, India). One hour after incubation in the dark, the absorbance of each reaction was measured at 650 nm (BioTek, Winooski, VT, USA). Percentage inhibition was calculated after subtracting the background and then compared to the negative control using the following formula:

$$\% \text{ inhibition of parasites} = 100 \text{ x} \frac{\text{OD negative control} - \text{OD sample}}{\text{OD negative control}}$$

### *In vitro* cytotoxicity assessment

Crude extracts were used to test toxicity at concentrations ranging from 3.125 to 800 μg/mL according to the protocol from previous studies [35, 36]. HepG2 and Vero cell lines at a density of $1.5 \times 10^4$ cells/well in 199 μL culture medium were seeded in a 96-well plate and then kept for 24 h at 37°C in a $CO_2$ incubator until they reached confluence. After 24 h incubation, both cell lines were added with different concentrations of the crude extract (1 μL). Doxorubicin (Sigma-Aldrich, New Delhi, India) chemotherapeutic drug, at a final concentration ranging from 0.3 to 20 μg/mL, was used as a positive control. DMSO and PBS were used as negative controls for the ethanolic and aqueous crude extracts. The treated cells were incubated at 37°C for 48 h. The experiment was performed in triplicate. After 48 h of incubation, tetrazolium salt (MTT) reagent (5 mg/mL) was added to each well and kept in an incubator at 37°C for 3 h. At the end of the exposure period, the supernatant reagent was discarded and replaced with 100 μL of DMSO. Cellular metabolic activity was determined by measuring the absorbance at a wavelength of 590 nm using a microplate reader. The percentage of cytotoxicity was calculated after subtracting the background and then compared to the negative control using the following formula:

$$\% \text{ cytotoxicity} = 100 - \left[ 100 \text{ x } \frac{\text{OD sample}}{\text{OD negative control}} \right]$$

The results of the cytotoxicity test are shown as half-maximal cytotoxicity concentration ($CC_{50}$), as determined by regression analysis using GraphPad Prism 6 program. The effectiveness of the extracts in inhibiting parasite growth was classified using the $IC_{50}$ value based on four categories: high, good, moderate, and no activity. High activity was defined as an $IC_{50}$ less than 5 μg/mL, good activity as 5–15 μg/mL, moderate activity as 15–50 μg/mL, and no activity as more than 50 μg/mL [37].

### Selectivity index

A selectivity index (SI), the ratio between the $CC_{50}$ of *in vitro* cytotoxicity and the $IC_{50}$ of *in vitro* antimalarial activity, was calculated for each crude extract using the following formula:

$$\text{SI} = \frac{CC_{50} \text{ of } \textit{in vitro} \text{cytotoxicity}}{IC_{50} \text{ of } \textit{in vitro} \text{antimalarial activity}}$$

### Gas chromatography-mass spectrometry analysis

The metabolite profiles of both CTLL and PSCD formulations were determined by gas chromatography (GC) using a 7000C Triple Quadrupole GC/mass spectrometry (MS) (Agilent Technologies, Santa Clara, CA, USA) equipped with an HP5MS column (30 m x 0.25 mm; 0.25 μm), according to previous studies [36, 38]. An electron ionization system that sets high-energy electrons at 70 eV, ion source temperature of 250°C, and mass scanning range of 33–600 amu in a full scan was used to identify spectroscopic detection. Pure helium gas (99.99% purity) was used as the carrier gas at a constant flow rate of 1 mL/min. The injector temperature was maintained at 250°C, and the oven temperature was set as follows: 60°C for 2 min, 150°C at a rate of 10°C/min, and 300°C at an increasing rate of 5°C/min. One microliter of the sample in ethanol was injected in the split mode at a split ratio of 20:1. Retention times and mass spectra were compared with those in the spectral database of the National Institute of Standards and Technology (NIST2011) structural library to identify the compounds in the test samples. Compounds were selected and identified from the peaks with 80% similarity in the database.

## Experimental animal

One hundred and fifty Male Institute of Cancer (ICR) adult mice aged 6–8 weeks weighing 20–30 g was obtained from Nomura Siam International Co., Ltd., Bangkok, Thailand. The animals were housed in groups of five per cage and acclimatized for 7 days in standard and constant laboratory conditions (22 ± 3°C, 50–60% humidity, and under 12–12 h light-dark cycles) with free access to a standard laboratory diet and clean water. The animal care staff handled sanitation by cleaning the cages and replacing bedding material every day [39]. The activity level and overall appearance of the animals were monitored at a minimum of two times a day during experimentation. Mice were strictly managed according to the international guidelines for animals used in experiments. *Plasmodium berghei* ANKA strain, a rodent malarial parasite, was provided by Thomas F. McCutchan and obtained from the Biodefense and Emerging Infections Research Resources Repository, National Institute of Allergy and Infectious Diseases, and National Institute of Health. Mouse donors were injected with *P. berghei*-infected RBCs via intraperitoneal to induce parasitic proliferation. When the mouse donors had parasitemia levels of 20–30%, blood samples were collected by cardiac puncture and stored in heparinized tubes for injection into the experimental mice. All surgical procedures were performed under isoflurane anesthesia in order to minimize the suffering of mice. Blood collected from the donor mice was diluted with 0.9% physiological isotonic saline. Each mouse used in the experiment was infected with 0.2 mL of infected RBCs containing approximately $1 \times 10^7$ *P. berghei*-parasitized RBCs. During the experimental period, the mice showing signs of severe, obvious pain, localized tissue damage, and markers associated with death or a poor prognosis of quality of life were eliminated to alleviate the duration of pain and distress by isoflurane euthanasia.

## Animal grouping and dosing

Infected male ICR mice were randomly divided into nine groups, with five mice each. Group 1 (infected control mice) was administered a mixture of 7% Tween 80 and 3% ethanol in distilled water. Groups 2 and 3 (positive controls) were administered 6 mg/kg body weight of artesunate and 25 mg/kg body weight of chloroquine, respectively. Groups 4, 5, and 6 were administered 200, 400, and 600 mg/kg body weight of the CTLL crude extract formulation, respectively. Groups 7, 8, and 9 were administered 200, 400, and 600 mg/kg body weight of the PSCD formulation crude extract, respectively. The doses of the crude extract were selected based on the median lethal dose ($LD_{50}$) value from the acute oral toxicity study and the preliminary results of the extracts. The volume administered was calculated based on the body weight of the individual mouse, and the treatment duration was based on the type of test performed. For oral acute toxicity testing, mice were divided into five groups, with five mice each. Group 1 (control group) received no treatment, group 2 (negative control group) was administered a mixture of 7% Tween 80 and 3% ethanol in distilled water, group 3 was administered a dose of 2,000 mg/kg body weight of the CTLL formulation crude extract, and group 4 was administered a dose of 2,000 mg/kg body weight of the PSCD formulation crude extract. Acute toxicity was induced in mice via oral administration.

## Four-day suppressive test (activity on early infection)

The 4-day suppressive test was performed to determine the schizonticidal activities of crude extracts on early infection with the *P. berghei* ANKA strain in ICR mice, as described by Knight and Peters [40]. In this test, the mice were randomly divided into nine groups, with five mice each, weighed, and maintained on a standard diet. Forty-five mice were inoculated with $1 \times 10^7$ *P. berghei*-infected RBCs via intraperitoneal injection. The crude extracts were

initiated for treatment 3 h after infection with the parasite on day 0 and continued for the next 4 days from day 0 to day 3, with a 24-h interval between doses. In the extract treatment group, mice were administered a dose of 200, 400, and 600 mg/kg body weight ethanolic extracts of CTLL and PSCD formulations in 200 μL of 7% Tween 80 daily. The infected group was administered 200 μL of 7% Tween 80, whereas the positive control group was administered 6 mg/kg body weight of artesunate or 25 mg/kg body weight of chloroquine orally daily. At the end of the treatment (11/07/2022), blood from the tail vein of each mouse was smeared onto a thin blood film to evaluate the percentage of parasitemia and suppression. The mice were given 2% isoflurane under inhalation anesthesia immediately after blood collection and then euthanized with a cardiac puncture.

### Curative/Rane's test (activity on established infection)

The curative test was performed with the crude extracts that presented the greatest parasitemia inhibition in the 4-day suppressive test. Curative potential against *P. berghei* ANKA infection in mice was determined according to the protocol described by Ryley and Peters [41]. Mice were intraperitoneally inoculated with $1 \times 10^7$ *P. berghei* ANKA-infected RBCs on the first day (day 0). Seventy-two hours post-infection, 45 mice were randomly divided into 9 groups, with 5 mice each, as described above. The mice in each group were treated daily for 5 days, starting from days 3 to 7 post-infection. On days 3 (10/07/2022) and day 8 post-infection (15/07/2022), Giemsa-stained thin blood films were examined to monitor the level of parasitemia. After collecting blood samples, 2% isoflurane was used for anesthesia, and cardiac puncture was used to euthanize immediately.

### Prophylactic/repository test (activity on residual infection)

The prophylactic potential against *P. berghei* ANKA infection in ICR mice was determined using residual infection according to the protocol described by Peters [42]. On the first day (day 0) of the test, 45 ICR mice were randomly grouped into 5 groups, with 5 mice each, and treated as described earlier. They were orally administered 200, 400, and 600 mg/kg of ethanolic extracts of the CTLL formulation, 200, 400, and 600 mg/kg body weight of ethanolic extracts of the PSCD formulation, 6 mg/kg body weight of artesunate, 25 mg/kg body weight of chloroquine, or 7% Tween 80 in distilled water for 3 consecutive days. On day 3, the mice were infected with a standard inoculum of $1 \times 10^7$ *P. berghei*-infected RBCs administered intraperitoneally. Seventy-two hours post-infection (07/07/2022), the percentage of parasitemia was evaluated by collecting blood from the tail vein of each mouse. Immediately, the mice received 2% isoflurane for anesthesia and were then euthanized with a cardiac puncture.

### Parasite determination

Blood samples were collected from the tail vein of each mouse on day 5 post-infection for a 4-day suppressive test, day 8 post-infection for a curative test, and days 3 and 8 post-infection for a prophylactic test. Smeared slides were dried for 5 min at room temperature, stained with Giemsa solution for 10 min, washed gently with distilled water, and air-dried at room temperature. The slides were examined under a light microscope using an oil-immersion objective at 100x magnification power, and an average of five fields were randomly selected. Parasite counts were determined by two independent observers who were blinded to the treatment. The percentage of blood parasites was calculated by counting parasitized RBCs and 300 RBCs from Giemsa-stained thin blood films, and the average percentage suppression of parasitemia was calculated by comparing the parasitemia in the infected group with that of the treated

mice using the following formula:

$$\% \text{ parasitemia} = \frac{\text{Number of parasitized red blood cells}}{\text{Total number of red blood cells counted}} X100$$

$$\% \text{ suppression} = \frac{A-B}{A} X\ 100$$

Where A is the mean parasitemia of the negative control group, and B is the mean parasitemia of the treated group.

## Acute oral toxicity test

The oral acute toxicity of ethanolic crude extracts of CTLL and PSCD formulations was investigated in male ICR mice in accordance with the standard guideline procedures of the Organization for Economic Cooperation and Development (OECD) [43]. Twenty ICR mice were randomly divided into four groups, with five mice each, as described in the animal grouping and dosing section. The mice were not allowed access to food or water 3 h before the treatment. After fasting, the mice in the treatment group were orally administered a single dose of 2,000 mg/kg body weight of CTLL and PSCD formulation extracts. The negative control group was administered a mixture of 7% Tween 80 and 3% ethanol in distilled water, while the control group received no treatment. Physical and behavioral changes in mice, including muscle power, emotion, sleep, excretion, feeding, and hair erection, were observed 3 h after treatment and continued daily for 14 days. Food and water consumption was recorded daily. Body weight changes were measured on days 0 and 14 using a sensitive digital weighing balance (Mettler Toledo, model: ML3002E, Indonesia). On day 14, mice were anesthetized with 2% isoflurane (Piramal Pharma, PA, USA) by inhalation using a rodent anesthesia machine, and the test was to ensure deep anesthesia using the toe pinch on each toe and corneal reflexes. After anesthetization, blood samples were collected by cardiac puncture for biochemical analyses. Liver and kidney tissues were immediately harvested to study the histopathological changes using hematoxylin and eosin staining (16/07/2022).

## Biochemical analysis

In the acute toxicity test, blood samples were collected in a heparinized tube using the cardiac puncture technique. To separate the plasma, blood was centrifuged at $3,000 \times$ g for 5 min. Liver and kidney functions were evaluated by measuring plasma biochemical parameters, including aspartate aminotransferase (AST), alanine aminotransferase (ALT), alkaline phosphate (ALP), blood urea nitrogen (BUN), and creatinine, using an AU 480 chemistry analyzer (Beckman Coulter, USA).

## Liver and kidney histopathological examination

Histopathological examinations were performed following previously described standard laboratory procedures [44, 45]. On day 14 of the acute oral toxicity test, 5 mice per group were anesthetized with 2% isoflurane (Piramal Pharma, PA, USA) by inhalation using rodent anesthesia machines and euthanized by cardiac puncture. Then, liver and kidney tissues were collected, labeled, and immediately fixed in 10% (v/v) formalin at 25°C. After fixation, the tissues were dehydrated in a series of alcohol concentrations (starting at 50%, 70%, 90%, and 100% ethanol), cleared with xylene solution, and embedded in paraffin. The paraffin-embedded tissues were sectioned at 5 μm thickness using a microtome and stained with hematoxylin and

eosin solution. The stained slides were evaluated under a light microscope by two independent observers blinded to the experimental groups.

## Data analysis

All data were accessed for analysis on the date of 20/07/2022 to 30/08/2022. Data analysis was performed using Statistical Package for the Social Sciences statistical software version 23 (IBM, Armonk, NY, USA). The results of the study were expressed as the mean and standard error of the mean (mean ± SEM). Normal distribution was evaluated using the Kolmogorov–Smirnov test. The comparison between parasitemia, percentage of suppression, food and water consumption, body weight, and liver and lung biochemical parameters among the groups was performed using a one-way analysis of variance descriptive statistics, followed by a post hoc Tukey's multiple comparison test. A $P < 0.05$ was considered statistically significant.

## Results

### Extraction yields

The percentage yields of CTLL and PSCD formulations and their plant ingredients are shown in Tables 2 and 3, respectively. Water extraction produced a higher percentage yield of CTLL and PSCD than that of ethanol extraction. Among the plant constituents used in the CTLL formulation, the water extract of *A. lancea* produced the highest percentage yield (49.42%), whereas the ethanolic extract of *E. longifolia* the lowest percentage yield (1.24%) (Table 2). Among the plant constituents in the PSCD formulation, the water extract of *Nelumbo nucifera* showed the highest percentage yield (37.23%), whereas the ethanolic extract of *Myristica fragrans* showed the lowest percentage yield (2.08%) (Table 3).

### *In vitro* antimalarial activity

Table 4 shows the *in vitro* antimalarial activities of extracts of the CTLL formulation and its plant ingredients. The results demonstrated that the ethanolic extract of the CTLL formulation (IC$_{50}$ = 4.88 μg/mL) presented high antimalarial activity, whereas the water extract of the CTLL formulation (IC$_{50}$ = 32.83 μg/mL) presented moderate antimalarial activity (Table 4). Among the plant ingredients in the CTLL formulation, high antimalarial activity was found in the aqueous extract of *E. longifolia* (IC$_{50}$ = 2.12 μg/mL). Good antimalarial activity was found in ethanolic extract of *A. lancea* (IC$_{50}$ = 7.37 μg/mL), *Acetosella vulgaris* (IC$_{50}$ = 9.37 μg/mL), and *Dracaena loureiroi* (IC$_{50}$ = 10.47 μg/mL). Three plant ingredients in the CTLL formulation

**Table 2. Percentage yield of Chan-Ta-Lee-La (CTLL) formulations and its plant ingredients.**

| Formulation/Plant species | Part used | % Yield | |
|---|---|---|---|
| | | **Ethanol** | **Water** |
| CTLL formulation | Powder | 6.63 | 14.42 |
| *Angelica sylvestris* | Roots | 4.18 | 25.42 |
| *Atracylodes lancea* | Rhizomes | 23.75 | 49.42 |
| *Artemisia vulgaris* | Whole plant | 6.63 | 27.57 |
| *Santalum album* | Stem | 4.05 | 6.75 |
| *Dracaena loureiroi* | Stem | 6.42 | 10.97 |
| *Gymnopetalum cochinense* | Fruits | 11.35 | 22.58 |
| *Tinospora crispa* | Stem | 6.06 | 11.48 |
| *Eurycoma longifolia* | Roots | 1.24 | 2.90 |

**Table 3. Percentage yield of Pra-Sa-Chan-Dang (PSCD) formulation and its plant ingredients.**

| Formulation/Plant species | Part used | % Yield | |
|---|---|---|---|
| | | Ethanol | Water |
| PSCD formulation | Powder | 10.61 | 11.14 |
| *Dracaena loureiroi* | Stem | 6.42 | 10.97 |
| *Heliciopsis terminalis* | Roots | 2.88 | 8.04 |
| *Bouea macrophylla* | Roots | 7.47 | 6.34 |
| *Citrus aurantifolia* | Roots | 3.19 | 5.01 |
| *Kaempferia galanga* | Rhizomes | 12.30 | 21.07 |
| *Ligusticum sinense* | Rhizomes | 4.12 | 33.57 |
| *Myristica fragrans* | Stem | 2.08 | 3.35 |
| *Caesalpinia sappan* | Stem | 8.77 | 8.64 |
| *Nelumbo nucifera* | Pollens | 22.41 | 37.23 |
| *Mesua ferrea* | Flower | 17.57 | 31.08 |
| *Mammea siamensis* | Flower | 12.55 | 21.08 |
| *Jasminum sambac* | Flower | 2.57 | 34.87 |

showed moderate antimalarial activity, including ethanolic extract of *A. sylvestris* ($IC_{50}$ = 28.10 μg/mL), *G. cochinense* ($IC_{50}$ = 35.72 μg/mL), and *T. crispa* ($IC_{50}$ = 43.17 μg/mL).

Table 5 shows the *in vitro* antimalarial activities of the PSCD formulation and its plant ingredients. The results demonstrated that the ethanolic extract of the PSCD formulation ($IC_{50}$ = 4.19 μg/mL) presented high antimalarial activity, whereas the water extract of the PSCD formulation ($IC_{50}$ = 59.19 μg/mL) presented no antimalarial activity (Table 5). Among

**Table 4. Half-maximal inhibitory concentration ($IC_{50}$), half-maximal cytotoxic concentration ($CC_{50}$), and selectivity index (SI) of CTLL formulation and its ingredients.**

| Formulation/Plant species | | pLDH | MTT ($CC_{50}$, μg/mL) | | SI ($CC_{50}/IC_{50}$) | |
|---|---|---|---|---|---|---|
| | | ($IC_{50}$, μg/mL) | Vero cells | HepG2 cells | Vero cells | HepG2 cells |
| CTLL formulation | Ethanol | 4.88 ± 0.36 | 199.40 ± 6.10 | 189.05 ± 6.95 | 40.86 | 38.74 |
| | Water | 32.83 ± 5.65 | 134.50 ± 12.50 | 289.75 ± 17.15 | 4.10 | 8.83 |
| *Angelica sylvestris* | Ethanol | 28.10 ± 0.82 | 158.10 ± 8.00 | 767.40 ± 7.80 | 5.63 | 27.31 |
| | Water | > 100 | > 800 | > 800 | < 8 | < 8 |
| *Atracylodes lancea* | Ethanol | 7.37 ± 7.72 | 110.55 ± 0.45 | 147.70 ± 5.70 | 15 | 20.04 |
| | Water | > 100 | > 800 | > 800 | < 8 | < 8 |
| *Artemisia vulgaris* | Ethanol | 9.37 ± 0.82 | 50.15 ± 0.02 | 29.31 ± 0.62 | 5.35 | 3.13 |
| | Water | > 100 | 714.35 ± 8.95 | > 800 | < 7.14 | < 8 |
| *Santalum album* | Ethanol | > 100 | 176.05 ± 9.05 | 566.40 ± 4.10 | < 1.76 | < 5.66 |
| | Water | 72.02 ± 1.53 | > 800 | > 800 | > 11.11 | > 11.11 |
| *Dracaena loureiroi* | Ethanol | 10.47 ± 0.00 | 77.41 ± 4.95 | 398.40 ± 11.70 | 7.39 | 38.05 |
| | Water | 104.00 ± 0.00 | 270.40 ± 19.3 | > 800 | 2.60 | > 7.69 |
| *Gymnopetalum cochinense* | Ethanol | 35.72 ± 3.29 | 57.24 ± 1.32 | 95.97 ± 1.14 | 1.60 | 2.68 |
| | Water | > 100 | 91.39 ± 6.63 | 333.10 ± 1.70 | < 0.91 | < 3.33 |
| *Tinospora crispa* | Ethanol | 43.17 ± 7.54 | 501.05 ± 3.65 | 201.40 ± 0.80 | 11.61 | 4.67 |
| | Water | 100.37 ± 8.53 | > 800 | 536.90 ± 25.80 | > 7.97 | 5.36 |
| *Eurycoma longifolia* | Ethanol | 99.15 ± 1.92 | 186.9 ± 0.15 | 34.88 ± 2.20 | 1.89 | 0.35 |
| | Water | 2.12 ± 0.13 | 74.01 ± 5.08 | 68.95 ± 4.87 | 34.91 | 32.52 |

CTLL, Chan-Ta-Lee-La; pLDH, *Plasmodium* lactate dehydrogenase; MTT, tetrazolium salt

**Table 5. Half-maximal inhibitory concentration ($IC_{50}$), half-maximal cytotoxic concentration ($CC_{50}$), and selectivity index (SI) of PSCD formulation and its ingredients.**

| Formulation/Plant species | | pLDH | MTT ($CC_{50}$, µg/mL) | | SI ($CC_{50}/IC_{50}$) | |
|---|---|---|---|---|---|---|
| | | ($IC_{50}$, µg/mL) | Vero cells | HepG2 cells | Vero cells | HepG2 cells |
| PSCD formulation | Ethanol | 4.19 ± 0.33 | 42.74 ± 1.45 | 118.70 ± 3.60 | 10.20 | 28.33 |
| | Water | 59.19 ± 4.90 | 209.55 ± 8.15 | 402.80 ± 16.50 | 3.54 | 6.80 |
| *Dracaena loureiroi* | Ethanol | 10.47 ± 0.00 | 77.41 ± 4.95 | 398.40 ± 11.70 | 7.39 | 38.05 |
| | Water | > 100 | 270.40 ±19.30 | > 800 | 2.60 | > 7.69 |
| *Heliciopsis terminalis* | Ethanol | 7.62 ± 0.74 | 89.75 ± 0.06 | 98.31 ± 2.38 | 11.78 | 12.90 |
| | Water | 49.38 ± 2.25 | 382.95 ± 14.65 | > 800 | 7.76 | > 16.20 |
| *Bouea macrophylla* | Ethanol | 15.48 ± 2.08 | 421.75 ± 1.75 | 87.37 ± 4.17 | 27.24 | 5.64 |
| | Water | 30.46 ± 7.10 | 644.55 ± 12.75 | 190.45 ± 15.35 | 21.16 | 6.25 |
| *Citrus aurantifolia* | Ethanol | 12.14 ± 3.66 | 110.65 ± 6.45 | 246.95 ± 10.75 | 9.11 | 20.34 |
| | Water | > 100 | > 800 | >800 | < 8.00 | <8.05 |
| *Kaempferia galanga* | Ethanol | 25.80 ± 0.88 | > 800 | 98.31 ± 2.38 | > 31.00 | 3.81 |
| | Water | > 100 | > 800 | > 800 | < 8.00 | < 8.00 |
| *Ligusticum sinense* | Ethanol | 10.41 ± 2.52 | 77.07 ± 3.54 | 116.85 ± 1.45 | 7.40 | 11.22 |
| | Water | > 100 | 320.50 ± 9.70 | > 800 | < 3.21 | <8.00 |
| *Myristica fragrans* | Ethanol | 8.87 ± 0.00 | 19.28 ± 0.91 | 42.27 ± 0.50 | 2.17 | 4.77 |
| | Water | 43.23 ± 4.43 | 685.80 ± 9.10 | > 800 | 15.86 | > 18.50 |
| *Caesalpinia sappan* | Ethanol | 6.24 ± 0.07 | 41.35 ± 0.67 | 53.57 ± 3.21 | 6.63 | 8.58 |
| | Water | 9.93 ± 1.39 | 44.73 ± 2.30 | 82.23 ± 9.86 | 4.50 | 8.28 |
| *Nelumbo nucifera* | Ethanol | 53.08 ± 2.48 | 58.78 ± 7.95 | 566.30 ± 13.30 | 1.11 | 10.67 |
| | Water | 157.9 ± 14.91 | 226.40 ± 20.00 | 609.95 ± 10.35 | 1.43 | 3.86 |
| *Mesua ferrea* | Ethanol | 4.30 ± 0.73 | 25.10 ± 1.99 | 131.80 ± 5.80 | 5.84 | 30.65 |
| | Water | > 100 | 290.80 ± 11.10 | > 800 | 2.59 | > 7.13 |
| *Mammea siamensis* | Ethanol | 0.99 ± 0.18 | 8.33 ± 0.04 | 15.36 ± 2.32 | 8.41 | 15.52 |
| | Water | 82.29 ± 2.17 | 525.35 ± 13.45 | > 800 | 6.38 | > 9.72 |
| *Jasminum sambac* | Ethanol | 26.06 ± 0.52 | 297.70 ± 5.70 | 312.50 ± 15.6 | 11.42 | 11.99 |
| | Water | > 100 | 212.65 ± 21.35 | > 800 | < 2.13 | < 8.00 |

pLDH, *Plasmodium* lactate dehydrogenase; MTT, tetrazolium salt

the plant ingredients in the PSCD formulations, high antimalarial activity was found in the ethanolic extract of *M. siamensis* ($IC_{50}$ = 0.99 µg/mL). The top three samples of plant ingredients with good antimalarial activity were found in ethanolic extracts of *M. ferrea* ($IC_{50}$ = 4.30 µg/mL), *C. sappan* ($IC_{50}$ = 6.24 µg/mL), and *H. terminalis* ($IC_{50}$ = 7.62 µg/mL).

### *In vitro* cytotoxicity and selectivity index

In this study, the lowest toxic effects are indicated using the $CC_{50}$, which is higher than that of the tested concentration of 800 µg/mL. For CTLL formulation and its plant ingredients, the water extracts of *A. sylvestris*, *A. lancea* and *Santalum album* exhibited $CC_{50}$ > 800 µg/mL in both cell lines. All the water extracts, except for *E. longifolia*, were safer than that of the ethanol extracts (Table 4). The extracts with the highest toxicity were the ethanolic extract of *A. vulgaris*. The extract exhibited $CC_{50}$ on Vero cells at 50.15 ± 0.02 µg/mL and 29.31 ± 0.62 µg/mL on HepG2 cells. For the PSCD formulation and its ingredients, 15 out of 56 extracts (34.09%) revealed $CC_{50}$ > 800 µg/mL. The water extracts of *C. aurantifolia* and *K. galangal* showed the highest $CC_{50}$ value in both cell lines, whereas the ethanolic extract of *M. siamensis* exhibited the lowest $CC_{50}$ at 8.33 ± 0.04 µg/mL (Table 5).

Three extracts in the CTLL formulation, including water extracts of *S. album* and *E. longifolia*, and ethanolic extracts of *A. lancea*, showed SI > 10 in both cell lines (Table 4), whereas SI values > 10 in the PSCD formulation were found in the ethanolic extracts of *H. terminalis*, *J. sambac*, and water extracts of *M. fragrans* (Table 5). The results showed that the ethanolic extracts of the CTLL and PSCD formulations had SI > 10 in both cell lines. Therefore, the ethanolic extracts of the CTLL and PSCD formulations were selected for further investigation of their *in vivo* antimalarial activity and toxicity in mice.

## GC-MS analysis of the ethanolic extract of the CTLL and PSCD formulations

GC-MS analysis of the ethanolic extract of the CTLL formulation is shown in Table 6. Forty-nine compounds were detected, including linderol (13.14%), isoborneol (7.88%), β-eudesol (6.52%), linoleic acid (5.93%), and oleic acid (3.85%). A chromatogram of the detected compounds is shown in Fig 1. GC-MS analysis of the ethanolic extract of the PSCD formulation is presented in Table 7. The extracts contained 31 compounds. The most abundant compounds were ethyl 4-methoxycinnamate (5.32%), 3-hydroxy-2-(4-hydroxy-3-methoxyphenyl)-4H-chromen-4-one (1.97%), flamenol (1.59%), oleic amide (1.37%), and linoleic acid (1.28%). A chromatogram of the ethanolic extract of the PSCD formulation is shown in Fig 2.

## Four-day suppressive test

The percentages of parasitemia and suppression are summarized in Table 8. The effects of the formulations revealed that both ethanolic extracts reduced the percentage of parasitemia in a dose-dependent manner, and mice treated with drugs or extracts at the tested doses exhibited significant suppressive effects ($P < 0.05$) compared with the negative group. The highest percent suppression was observed in mice treated with artesunate and chloroquine. The percent suppression of the ethanolic extract of the CTLL formulation at a dose of 200, 400, and 600 mg/kg was 71.97 ± 2.96, 77.69 ± 2.53, and 89.80 ± 2.96, whereas the percent suppression of the ethanolic extract of the PSCD formulation was 70.40 ± 1.08, 73.65 ± 2.08, and 78.36 ± 3.46, respectively. However, only 600 mg/kg of the ethanolic extract of the CTLL formulation demonstrated significant suppression ($P < 0.05$) compared with the extract-treated groups.

## Curative test

The curative potential of the extracts is listed in Table 9. All groups treated with the extracts and drugs showed a significant curative effect compared with the negative control ($P < 0.05$). The highest parasitemia suppression was observed in the group receiving chloroquine treatment, and the drug eradicated parasite infection on day 8. The percent suppression in the artesunate-treated group was 83.99%. For mice treated with the extracts, the highest percent suppression was observed in the group treated with 600 mg/kg of the ethanolic extract of the CTLL formulation, and the extract at this dose significantly suppressed parasitemia, compared with 200 mg/kg of the ethanolic extract of the CTLL formulation and 200 and 400 mg/kg of the ethanolic extract of the PSCD formulation.

## Prophylactic test

The protective effects of the ethanolic extracts of the CTLL and PSCD formulations are shown in Table 10. The drug with the highest percent suppression was chloroquine (66.02 ± 3.25%), whereas artesunate exhibited lower protective effects than that of chloroquine, with a percent suppression of 28.78 ± 1.05. The effects of the extracts exhibited a dose-dependent trend. The

**Table 6. List of compounds in the ethanolic extract of the CTLL formulation analyzed by GC-MS.**

| No. | RT | Name of compound | Molecular formula | MW | Peak area (%) |
|---|---|---|---|---|---|
| 1 | 7.725 | tran-3,3-Dimethylcyclohexylideneethanal | $C_{10}H_{16}O$ | 152 | 0.03 |
| 2 | 7.774 | Bicyclo[2.2.1]heptan-2-ol, 1,5,5-trimethyl- | $C_{10}H_{18}O$ | 154 | 0.13 |
| 3 | 7.847 | Fenchol | $C_{10}H_{18}O$ | 154 | 0.17 |
| 4 | 8.292 | Pyranone | $C_6H_8O_4$ | 144 | 0.07 |
| 5 | 8.359 | (+)-Camphor | $C_{10}H_{16}O$ | 152 | 1.52 |
| 6 | 8.549 | Isoborneol | $C_{10}H_{18}O$ | 154 | 7.88 |
| 7 | 8.699 | Linderol | $C_{10}H_{18}O$ | 154 | 13.14 |
| 8 | 8.751 | Isomentol | $C_{10}H_{20}O$ | 156 | 0.13 |
| 9 | 8.840 | 4-Terpinenol | $C_{10}H_{18}O$ | 154 | 0.02 |
| 10 | 9.079 | 1,7,7-trimethylbicyclo[2.2.1]heptan-2-ol | $C_{10}H_{18}O$ | 154 | 0.25 |
| 11 | 9.144 | Isopulegol | $C_{10}H_{18}O$ | 154 | 0.04 |
| 12 | 9.603 | Isobornyl acetate | $C_{12}H_{20}O_2$ | 196 | 0.05 |
| 13 | 10.419 | Acetic acid, 1,7,7-trimethyl-bicyclo[2.2.1]hept-2-yl ester | $C_{12}H_{20}O_2$ | 196 | 0.04 |
| 14 | 11.375 | m-Eugenol | $C_{10}H_{12}O_2$ | 164 | 0.46 |
| 15 | 12.392 | Caryophyllene | $C_{15}H_{24}$ | 204 | 0.21 |
| 16 | 13.254 | Curcumene | $C_{15}H_{22}$ | 202 | 0.08 |
| 17 | 13.422 | a-Acorenol | $C_{15}H_{26}O$ | 222 | 0.12 |
| 18 | 13.808 | Ethylparaben | $C_9H_{10}O_3$ | 166 | 0.12 |
| 19 | 13.919 | 7-Epi-cis-sesquisabinene hydrate | $C_{15}H_{26}O$ | 222 | 0.06 |
| 20 | 14.191 | β-Guaiene | $C_{15}H_{24}$ | 204 | 0.07 |
| 21 | 14.366 | a-Acorenol | $C_{15}H_{26}O$ | 222 | 0.11 |
| 22 | 14.905 | Spathulenol | $C_{15}H_{24}O$ | 220 | 0.09 |
| 23 | 15.788 | 8-epi-gamma-Eudesmol | $C_{15}H_{26}O$ | 222 | 0.22 |
| 24 | 15.947 | Agarospirol | $C_{15}H_{26}O$ | 222 | 1.19 |
| 25 | 16.171 | β-Eudesmol | $C_{15}H_{26}O$ | 222 | 6.52 |
| 26 | 16.413 | Elemol | $C_{15}H_{26}O$ | 222 | 0.16 |
| 27 | 16.530 | β-Santalol | $C_{15}H_{24}O$ | 220 | 0.10 |
| 28 | 16.674 | a-Bisabolol | $C_{15}H_{26}O$ | 222 | 0.13 |
| 29 | 17.388 | trans-Nuciferol | $C_{15}H_{22}O$ | 218 | 0.26 |
| 30 | 17.535 | 4-((1E)-3-hydroxy-1-propenyl)-2-methoxyphenol | $C_{10}H_{12}O_3$ | 180 | 0.27 |
| 31 | 17.615 | 2(1H)naphthalenone, 3,5,6,7,8,8a-hexahydro-4,8a-dimethyl-6-(1-methylethenyl)- | $C_{15}H_{22}O$ | 218 | 0.20 |
| 32 | 19.276 | 3-isopropyl-6,7-dimethyltricyclo[4.4.0.0(2,8)]decane-9,10-diol | $C_{15}H_{26}O_2$ | 238 | 0.11 |
| 33 | 20.854 | 11-isopropylidenetricyclo[4.3.1.1(2,5)]undec-3-en-10-one | $C_{14}H_{18}O$ | | 0.19 |
| 34 | 21.035 | Diepicedrene-1-oxide | $C_{15}H_{24}O$ | 220 | 0.18 |
| 35 | 21.388 | β-santalol acetate | $C_{17}H_{26}O_2$ | 262 | 0.13 |
| 36 | 21.547 | n-hexadecanoic acid | $C_{16}H_{32}O_2$ | 256 | 3.49 |
| 37 | 22.126 | Ethyl palmitate | $C_{18}H_{36}O_2$ | 284 | 0.73 |
| 38 | 22.623 | 1,4-hexadien-3-one, 5-methyl-1-[2,6,6-trimethyl-2,4-cyclohexadien-1-yl]- | $C_{16}H_{22}O$ | 230 | 1.04 |
| 39 | 23.432 | Eudesma-5,11(13)-dien-8,12-olide | $C_{15}H_{20}O_2$ | 232 | 0.27 |
| 40 | 24.017 | Confertin | $C_{15}H_{20}O_3$ | 248 | 0.56 |
| 41 | 24.636 | Linoleic acid | $C_{18}H_{32}O_2$ | 280 | 5.93 |
| 42 | 24.722 | Oleic acid | $C_{18}H_{34}O_2$ | 282 | 3.85 |
| 43 | 25.096 | Octadecanoic acid | $C_{18}H_{36}O_2$ | 284 | 2.89 |
| 44 | 25.204 | Ethyl oleate | $C_{20}H_{38}O_2$ | 310 | 0.62 |
| 45 | 26.098 | 2-(3,7-dimethyl-octa-2,6-dienyl)-4-methoxy-phenol | $C_{17}H_{24}O_2$ | 260 | 0.11 |
| 46 | 27.217 | 5,8,11-heptadecatrienoic acid, methyl ester | $C_{18}H_{30}O_2$ | 278 | 0.13 |
| 47 | 27.469 | Ethyl linolenate | $C_{20}H_{34}O_2$ | 306 | 0.20 |

*(Continued)*

**Table 6.** (Continued)

| No. | RT | Name of compound | Molecular formula | MW | Peak area (%) |
|---|---|---|---|---|---|
| 48 | 28.410 | Oleamide | $C_{18}H_{35}NO$ | 281 | 1.37 |
| 49 | 28.501 | Oleamide | $C_{18}H_{35}NO$ | 281 | 0.34 |
| 50 | 28.581 | Oleamide | $C_{18}H_{35}NO$ | 281 | 0.33 |
| 51 | 29.032 | Pinostrobin chalcone | $C_{16}H_{14}O_4$ | 270 | 0.34 |

CTLL, Chan-Ta-Lee-La; GC-MS, gas chromatography-mass spectrometry

percent suppression at a dose of 200, 400, and 600 mg/kg of the ethanolic extract of the CTLL formulations were 0.41 ± 2.06, 0.92 ± 4.21, and 24.59 ± 4.33, and 3.27 ± 2.30, 57.45 ± 0.92, and 65.82 ± 2.23 for the PSCD formulation, respectively. However, parasitemia suppression was observed at a dose of 600 mg/kg of the CTLL formulation, and the 400 and 600 mg/kg CTLL formulations were significantly ($P < 0.05$) lower than that in the negative control group. Among all the extract-treated groups, the 600 mg/kg ethanolic extract of the PSCD formulation showed the highest percentage of parasitemia suppression, and there was no significant difference compared with chloroquine.

## Physical, behavioral, and body weight changes of oral acute toxicity test

Oral administration of 2,000 mg/kg of the ethanolic extract of the CTLL or PSCD formulation revealed that administration of a high dose did not produce any signs of toxicity. The mice did not show any alterations in their physical or behavioral characteristics throughout the experimental period. No alterations in body movements, excretion, feeding, or abnormal behaviors were observed. No mortality occurred within 14 days. For the effects of the extracts on body weight changes, the results showed that there was no statistically significant difference ($P < 0.05$) between the extracts, vehicle, and normal groups (Table 11).

## Biochemical tests

The effects of the ethanolic extracts of the CTLL and PSCD formulations on liver and kidney functions are shown in Table 12. Kidney parameters, including BUN and creatinine levels, were not significantly different ($P < 0.05$) among all groups. For kidney parameters, the AST of mice treated with 7% Tween 80 and 3% ethanol in distilled water was significantly higher

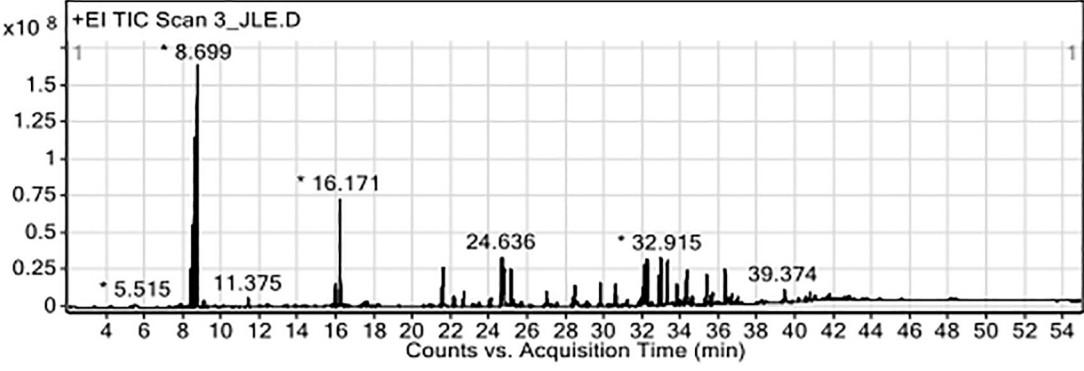

**Fig 1. Chromatogram of ethanolic extract of CTLL formulation from GC-MS analysis.** RT, retention time; MW, molecular weight; CTLL, Chan-Ta-Lee-La; GC-MS, gas chromatography-mass spectrometry.

**Table 7. List of compounds in the ethanolic extract of the PSCD formulation analyzed by GC-MS.**

| No. | RT | Name of compound | Molecular formula | MW | Peak area (%) |
|---|---|---|---|---|---|
| 1 | 8.264 | Pyranone | $C_6H_8O_4$ | 144 | 0.14 |
| 2 | 8.669 | L-borneol | $C_{10}H_{18}O$ | 154 | 0.04 |
| 3 | 8.834 | 4-Terpinenol | $C_{10}H_{18}O$ | 154 | 0.03 |
| 4 | 9.349 | Coumaran | $C_8H_8O$ | 120 | 0.27 |
| 5 | 10.394 | Anethole | $C_{10}H_{12}O$ | 148 | 0.08 |
| 6 | 10.783 | 4-hydroxy-3-methylacetophenone | $C_9H_{10}O_2$ | 150 | 0.06 |
| 7 | 12.248 | Cinnamic acid | $C_9H_8O_2$ | 148 | 0.06 |
| 8 | 12.515 | 2-acetylphenol | $C_8H_8O_2$ | 136 | 0.60 |
| 9 | 12.975 | Ethyl cinnamate | $C_{11}H_{12}O_2$ | 176 | 1.16 |
| 10 | 13.416 | Pentadecane | $C_{15}H_{32}$ | 212 | 0.95 |
| 11 | 13.839 | Flamenol | $C_7H_8O_3$ | 140 | 1.59 |
| 12 | 16.849 | (E)-p-methoxy-cinnamic acid | $C_{10}H_{10}O_3$ | 178 | 0.15 |
| 13 | 17.364 | p-heptylphenol | $C_{13}H_{20}O$ | 192 | 1.08 |
| 14 | 17.544 | 4-((1E)-3-hydroxy-1-propenyl)-2-methoxyphenol | $C_{10}H_{12}O_3$ | 180 | 0.12 |
| 15 | 17.909 | Ethyl 4-methoxycinnamate | $C_{12}H_{14}O_3$ | 206 | 5.32 |
| 16 | 18.035 | Lanceol, cis | $C_{15}H_{24}O$ | 220 | 0.24 |
| 17 | 18.841 | Syringic acid | $C_9H_{10}O_5$ | 198 | 0.08 |
| 18 | 21.035 | 1,4-methanoazulene-9-methanol, decahydro-4,8,8-trimethyl-, [1S-(1α,3aβ,4α,8aβ,9R*)]- | $C_{15}H_{26}O$ | 220 | 0.06 |
| 19 | 21.513 | Palmitic acid | $C_{16}H_{32}O_2$ | 256 | 1.05 |
| 20 | 22.129 | Ethyl palmitate | $C_{18}H_{36}O_2$ | 284 | 0.70 |
| 21 | 23.487 | Linolenic acid | $C_{18}H_{30}O_2$ | 278 | 0.93 |
| 22 | 24.017 | 6,9,12,15-docosatetraenoic acid, methyl ester | $C_{23}H_{38}O_2$ | 346 | 0.22 |
| 23 | 24.585 | Linoleic acid | $C_{18}H_{32}O_2$ | 280 | 1.28 |
| 24 | 24.680 | Oleic acid | $C_{18}H_{34}O_2$ | 282 | 1.17 |
| 25 | 25.057 | Octadecanoic acid | $C_{18}H_{36}O_2$ | 284 | 0.12 |
| 26 | 25.096 | Linoleic acid ethyl ester | $C_{20}H_{36}O_2$ | 308 | 0.84 |
| 27 | 25.201 | Ethyl oleate | $C_{20}H_{38}O_2$ | 310 | 0.45 |
| 28 | 25.375 | Palmitic acid amide | $C_{16}H_{33}NO$ | 255 | 0.17 |
| 29 | 26.332 | Pinosylvin, dimethyl ether | $C_{16}H_{16}O_2$ | 240 | 0.17 |
| 30 | 28.407 | Oleic acid amide | $C_{18}H_{35}NO$ | 281 | 1.37 |
| 31 | 34.960 | 3-hydroxy-2-(4-hydroxy-3-methoxyphenyl)-4H-chromen-4-one | $C_{16}H_{12}O_5$ | 284 | 1.97 |

RT, retention time; MW, molecular weight; PSCD, Pra-Sa-Chan-Dang; GC-MS, gas chromatography-mass spectrometry

than that of the normal control and the ethanolic extract of the PSCD formulation, which showed statistical significance ($P < 0.05$). ALP in mice treated with the ethanolic extract of the CTLL formulation was significantly ($P < 0.05$) higher than that in the normal and negative controls, whereas ALT showed no significant difference ($P < 0.05$) among all groups.

## Histopathological changes

The effects of the ethanolic extracts of the CTLL and PSCD formulations on histopathological changes in the liver and kidney are shown in Fig 3. Normal histology of the liver and kidneys is shown in Fig 3A and 3E. Histological analysis of the mice treated with the vehicle and extracts revealed that the kidney (3B, 3C, and 3D) and liver (3F, 3G, and 3H) sections appeared normal. The liver sections showed normal hepatocyte architecture without any signs of sinusoidal alterations or inflammatory cell infiltration. The extracts or vehicle did not cause acute kidney damage, as indicated by the normal structure of the glomeruli and renal tubules without the invasion of inflammatory cells.

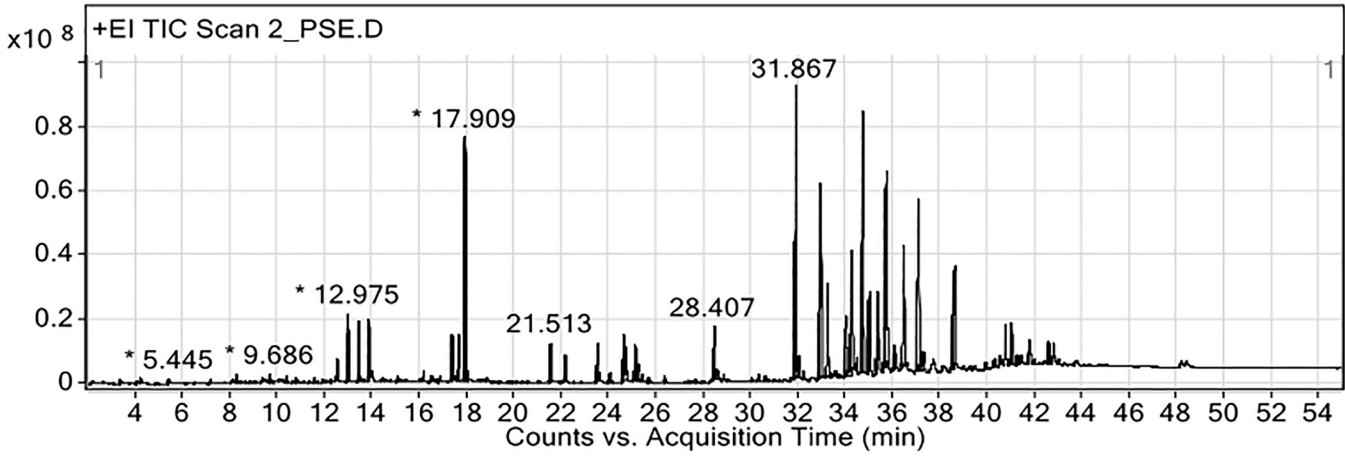

**Fig 2. Chromatogram of ethanolic extract of PSCD formulation from GC-MS analysis.** PSCD, Pra-Sa-Chan-Dang; gas chromatography-mass spectrometry.

## Discussion

Drug resistance has become one of the most important problems in infectious diseases, including malaria. Standard artemisinin resistance has emerged, resulting in delayed parasitic elimination after artemisinin-based combination therapies [46]. Therefore, the search for novel antimalarial agents has been urgently required [47]. According to the World Health Organization, traditional medicine strategies are widespread worldwide and have continued to increase each year since 2014 [48]. Approximately half of all the drugs currently in clinical use are derived from plants [49]. Moreover, plant materials are among the most important sources of

**Table 8. Suppressive effects of the ethanolic extract of the CTLL and PSCD formulations in 4-day suppressive test.**

| Treatment | Dose (mg/kg) | (% Parasitemia) | (% Suppression) |
|---|---|---|---|
| 7% Tween 80, 3% Ethanol | - | 19.82 ± 0.73 | - |
| Artesunate | 6 | 0 | 100 [a, d, e, g, h, i] |
| Chloroquine | 25 | 0 | 100 [a, d, e, g, h, i] |
| CTLL formulation | 200 | 5.56 ± 0.58 | 71.97 ± 2.96 [a, b, c, f] |
| | 400 | 4.42 ± 0.50 | 77.69 ± 2.53 [a, b, c, f] |
| | 600 | 2.02 ± 0.59 | 89.80 ± 2.96 [a, b, c, d, e, g, h, i] |
| PSCD formulation | 200 | 5.87 ± 0.21 | 70.40 ± 1.08 [a, b, c, f] |
| | 400 | 5.22 ± 0.41 | 73.65 ± 2.08 [a, b, c, f] |
| | 600 | 4.29 ± 0.69 | 78.36 ± 3.46 [a, b, c, f] |

Data are presented as the means ± SEM (n = 5 per group).

[a] Compared with mice receiving a mixture of 7% Tween 80 and 3% ethanol in distilled water as the negative control group

[b] compared with artesunate as the positive control group

[c] compared with chloroquine as the positive control group

[d] Compared with CTLL formulation at 200 mg/kg

[e] Compared with CTLL formulation at 400 mg/kg

[f] Compared with CTLL formulation at 600 mg/kg

[g] Compared with PSCD formulation at 200 mg/kg

[h] Compared with PSCD formulation at 400 mg/kg

[i] Compared with PSCD formulation at 600 mg/kg.

SEM, standard error of the mean; CTLL, Chan-Ta-Lee-La; PSCD, Pra-Sa-Chan-Dang

**Table 9. Percent suppression of the ethanolic extract of the CTLL and PSCD formulations in curative test.**

| Treatment | Dose (mg/kg) | (% Parasitemia) | (% Suppression) |
|---|---|---|---|
| 7% Tween 80, 3% Ethanol | - | 31.78 ± 0.63 | - |
| Artesunate | 6 | 5.09 ± 0.13 | 83.99 ± 0.40 [a, c, d, e, f, g, h, i] |
| Chloroquine | 25 | 0 | 100 [a, b, d, e, f, g, h, i] |
| CTLL formulation | 200 | 26.84 ± 0.30 | 15.45 ± 2.25 [a, b, c, e, f, i] |
|  | 400 | 25.49 ± 0.30 | 31.96 ± 1.17 [a, b, c, d, g, h] |
|  | 600 | 22.36 ± 0.55 | 35.94 ± 1.07 [a, b, c, d, g, h] |
| PSCD formulation | 200 | 26.87 ± 0.72 | 15.52 ± 0.93 [a, b, c, e, f, i] |
|  | 400 | 21.62 ± 0.37 | 19.79 ± 0.94 [a, b, c, e, f, i] |
|  | 600 | 20.36 ± 0.34 | 29.65 ± 1.74 [a, b, c, d, g, h, i] |

Data are presented as the means ± SEM (n = 5 per group)

[a] Compared with mice receiving a mixture of 7% Tween 80 and 3% ethanol in distilled water as the negative control group

[b] compared with artesunate as the positive control group

[c] compared with chloroquine as the positive control group

[d] Compared with CTLL formulation at 200 mg/kg

[e] Compared with CTLL formulation at 400 mg/kg

[f] Compared with CTLL formulation at 600 mg/kg

[g] Compared with PSCD formulation at 200 mg/kg

[h] Compared with PSCD formulation at 400 mg/kg

[i] Compared with PSCD formulation at 600 mg/kg.

SEM, standard error of the mean; CTLL, Chan-Ta-Lee-La; PSCD, Pra-Sa-Chan-Dang

**Table 10. Percent suppression of the ethanolic extract of the CTLL and PSCD formulations in prophylactic test on day 6.**

| Treatment | Dose (mg/kg) | (% Parasitemia) | (% Suppression) |
|---|---|---|---|
| 7% Tween 80, 3% Ethanol | - | 21.78 ± 0.70 | - |
| Artesunate | 6 | 15.51 ± 0.23 | 28.78 ± 1.05 [a, c, d, e, g, h, i] |
| Chloroquine | 25 | 7.40 ± 0.71 | 66.02 ± 3.25 [a, b, d, e, f, g] |
| CTLL formulation | 200 | 21.69 ± 0.45 | 0.41 ± 2.06 [b, c, f, h, i] |
|  | 400 | 21.58 ± 0.92 | 0.92 ± 4.21 [b, c, f, h, i] |
|  | 600 | 16.42 ± 0.94 | 24.59 ± 4.33 [a, c, d, e, g, h, i] |
| PSCD formulation | 200 | 21.07 ± 0.50 | 3.27 ± 2.30 [b, c, f, h, i] |
|  | 400 | 9.27 ± 0.45 | 57.45 ± 0.92 [a, b, d, e, f, g] |
|  | 600 | 7.44 ± 0.49 | 65.82 ± 2.23 [a, b, d, e, f, g] |

Data are presented as the means ± SEM (n = 5 per group)

[a] Compared with mice receiving a mixture of 7% Tween 80 and 3% ethanol in distilled water as the negative control group

[b] compared with artesunate as the positive control group

[c] compared with chloroquine as the positive control group

[d] Compared with CTLL formulation at 200 mg/kg

[e] Compared with CTLL formulation at 400 mg/kg

[f] Compared with CTLL formulation at 600 mg/kg

[g] Compared with PSCD formulation at 200 mg/kg

[h] Compared with PSCD formulation at 400 mg/kg

[i] Compared with PSCD formulation at 600 mg/kg.

SEM, standard error of the mean; CTLL, Chan-Ta-Lee-La; PSCD, Pra-Sa-Chan-Dang

**Table 11. Bodyweight of mice in acute toxicity test.**

| Group | Body weight (g) | | % Change |
|---|---|---|---|
| | **Day 0** | **Day 14** | |
| Normal mice | 35.97 ± 1.27 | 41.07 ± 1.65 | 14.13 ± 1.67 |
| 7% Tween 80, 3% Ethanol | 36.61 ± 0.35 | 40.12 ± 0.85 | 9.56 ± 1.74 |
| CTLL formulation | 35.55 ± 0.43 | 40.30 ± 0.79 | 13.36 ± 1.67 |
| PSCD formulation | 36.48 ± 0.49 | 41.46 ± 0.20 | 13.77 ± 2.09 |

Data are presented as the means ± SEM (n = 5 per group).

No statistically significant difference between the groups.

SEM, standard error of the mean; CTLL, Chan-Ta-Lee-La; PSCD, Pra-Sa-Chan-Dang

active compounds, and many previous studies have focused on the use of plants to search for new substances to cope with antimalarial drug problems [34]. They probably attack the parasite directly, causing minimal host damage [50]. Therefore, this study focused on traditional Thai formulations to stimulate the development of a new effective antimalarial agent. This study aimed to determine the anti-plasmodial activity and toxic effects of CTLL and PSCD formulations and their plant ingredients, which provide scientific justification for the treatment of many types of fever, including malaria-like symptoms [14, 15].

Our study on *in vitro* antimalarial activity demonstrated that among the 18-crude extracts of the CTLL formulation, the aqueous root extracts of *E. longifolia* show the highest antimalarial with the lowest $IC_{50}$ value of 2.12 μg/mL and $CC_{50}$ value of 74.01 μg/mL for Vero cells and 68.95 μg/mL for HepG2 cells, followed by the ethanolic extract of the CTLL formulation with $IC_{50}$ value of 4.88 μg/mL and $CC_{50}$ value of 199.40 μg/mL for Vero cells and 189.05 μg/mL for HepG2 cells, followed by the ethanolic rhizome extracts of *A. lancea* with $IC_{50}$ value of 7.37 μg/mL and $CC_{50}$ value of 110.55 μg/mL for Vero cells and 147.70 μg/mL for HepG2 cells, followed by the ethanolic rhizome extracts of *A. vulgaris* with $IC_{50}$ value of 9.37 μg/mL and $CC_{50}$ value of 50.15 μg/mL for Vero cells and 29.31 μg/mL for HepG2 cells, and followed by the ethanolic rhizome extracts of *D. loureiri* with $IC_{50}$ value of 10.47 μg/mL and $CC_{50}$ value of 77.41 μg/mL for Vero cells and 398.40 μg/mL for HepG2 cells. *E. longifolia* possesses many medicinal

**Table 12. Liver and kidney functions in acute toxicity test.**

| Parameters | Normal mice | 7% Tween 80, 3% Ethanol | CTLL formulation | PSCD formulation |
|---|---|---|---|---|
| **Liver function test** | | | | |
| AST (U/L) | 82.40 ± 1.69 [b] | 104.20 ± 6.81[a] | 88.60 ± 9.30 | 80.80 ± 1.11[b] |
| ALT (U/L) | 23.00 ± 0.71 | 24.20 ± 1.69 | 24.80 ± 1.36 | 24.40 ± 0.75 |
| ALP (U/L) | 83.00 ± 5.18 [c] | 83.60 ± 4.75 [c] | 119.40 ± 10.58 [a, b] | 102.00 ± 9.10 |
| **Kidney function test** | | | | |
| BUN (mg/dL) | 24.40 ± 1.21 | 25.20 ± 0.37 | 22.00 ± 0.84 | 22.00 ± 1.30 |
| Creatinine (mg/dL) | 0.16 ± 0.01 | 0.17 ± 0.01 | 0.15 ± 0.01 | 0.18 ± 0.01 |

Data are presented as the means ± SEM (n = 5 per group).

[a] Compared with normal mice

[b] Compared with a mixture of 7% Tween 80 and 3% ethanol in distilled water as the negative control group

[c] Compared with 2,000 mg/kg CTLL formulation

[d] Compared with 2,000 mg/kg PSCD formulation.

SEM, standard error of the mean; CTLL, Chan-Ta-Lee-La; PSCD, Pra-Sa-Chan-Dang; AST, aspartate aminotransferase; ALT, alanine aminotransferase; ALP, alkaline phosphatase; BUN, blood urea nitrogen

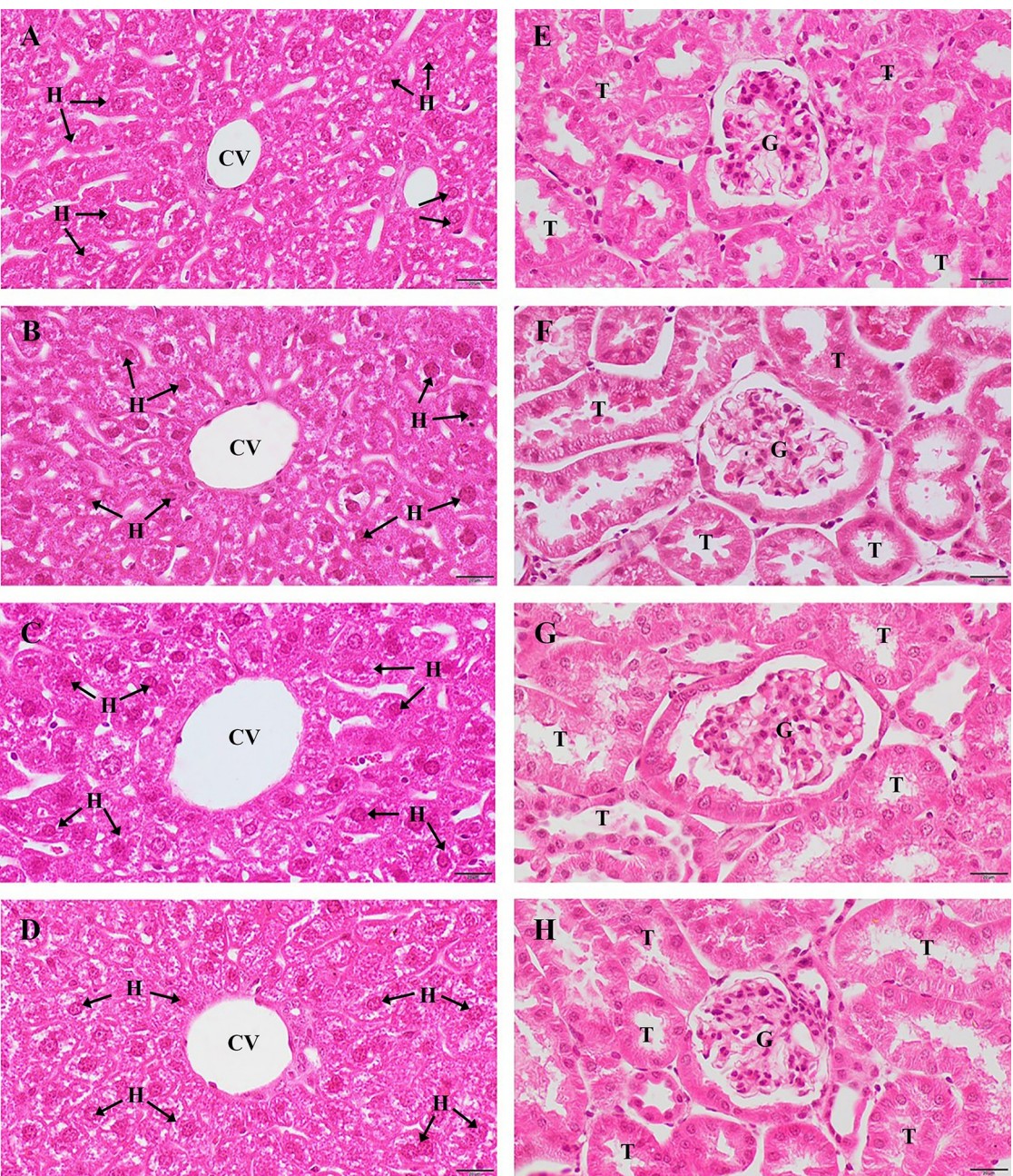

**Fig 3. Histopathological alteration of liver and kidney tissues.** (A) liver section of normal mice, (B) liver section of the negative control group, (C) liver section of the 2,000 mg/kg CTLL group, (D) liver section of the 2,000 mg/kg PSCD group, (E) kidney section of normal mice, (F). kidney section of the negative control group, (G) kidney section of the 2,000 mg/kg CTLL group, and (H) kidney section of the 2,000 mg/kg PSCD group. All images are 20x magnification. Bar = 100 μm. Central vein (CV), hepatocyte (H), tubule (T), and glomerulus (G); CTLL, Chan-Ta-Lee-La; PSCD, Pra-Sa-Chan-Dang.

properties, including malaria treatment, and the root extract of this plant shows a slight antimalarial effect in *Plasmodium yoelii*-infected mice [51]. *A. lancea* has been reported to exhibit antimalarial effects with 60% suppression at a dose of 400 mg/kg body weight [18]. The ethanolic leaf extract of *A. vulgaris* also inhibited parasitemia levels in a 4-day suppression assay [52]. Moreover, *D. loureirin* exhibited antimalarial properties against chloroquine-resistant

*P. falciparum* [30]. Based on the aforementioned extracts, the ethanolic extract of the CTLL formulation presents high antimalarial activity. In addition, the toxic effect was consistent with previous reports that the extract of the CTLL formulation did not produce signs of acute and sub-chronic toxicity in mice [53]. Therefore, the ethanolic extract of the CTLL formulation was selected for further *in vivo* investigation.

For the 26-crude extracts of the PSCD formulation, the ethanolic flower extracts of *M. siamensis* show the highest antimalarial with the lowest $IC_{50}$ value of 0.99 μg/mL and $CC_{50}$ value of 8.33 μg/mL for Vero cells and 15.36 μg/mL for HepG2 cells, followed by the ethanolic flower extracts of *M. ferrea* with $IC_{50}$ value of 4.30 μg/mL and $CC_{50}$ value of 25.10 μg/mL for Vero cells and 131.80 μg/mL for HepG2 cells, followed by the ethanolic extract of the PSCD formulation with $IC_{50}$ value of 4.19 μg/mL and $CC_{50}$ value of 42.74 μg/mL for Vero cells and 118.70 μg/mL for HepG2 cells, followed by the ethanolic stem extracts of *C. sappan* with $IC_{50}$ value of 6.24 μg/mL and $CC_{50}$ value of 41.35 μg/mL for Vero cells and 53.37 μg/mL for HepG2 cells, followed by the ethanolic rhizome extracts of *H. terminalis* with $IC_{50}$ value of 7.62 μg/mL and $CC_{50}$ value of 89.75 μg/mL for Vero cells and 98.31 μg/mL for HepG2 cells, and followed by the ethanolic stem extracts of *M. fragrans* with $IC_{50}$ value of 8.87 μg/mL and $CC_{50}$ value of 19.28 μg/mL for Vero cells and 42.27 μg/mL for HepG2 cells. The results from this study are in agreement with the previous study. The isolated compound from *M. siamensis* exhibited the most potent antimalarial activity with an $IC_{50}$ value of 9.57 μM and had the best binding affinity to the pLDH enzyme [29]. *M. ferrea* exhibits antimalarial properties against chloroquine-resistant *P. falciparum* [30]. For *C. sappan*, *Cassane diterpenes* isolated compounds from this plant were shown to have antimalarial properties against a chloroquine-resistant K1 strain of *P. falciparum* with $IC_{50}$ values of 0.78 μM [54]. In addition, *M. fragrans* exhibited antimalarial effects with 50% suppression at a dose of 600 mg/kg body weight in a 4-day suppressive test, with no toxic effects in an acute toxicity model [18]. In this study, the ethanolic extract of PSCD showed high antimalarial activity. This toxic effect is in line with a previous study that demonstrated the safety profile of the PSCD formulation. In this study, it was found that the PSCD formulation demonstrated a normal range of clinical biochemistry of liver and kidney function, histopathology, and concentrations of CYP2E1 [55]. Therefore, the ethanolic extract of the PSCD formulation was selected for *in vivo* analysis.

The SI value, calculated from the ratio between the toxic concentration in human cells ($CC_{50}$) and the effective concentration to prevent parasite growth ($IC_{50}$), is a parameter for evaluating whether additional studies on the extract are warranted [56]. The extract is considered potentially safe regarding cytotoxicity when the SI value is > 10 [57]. In this study, the ethanolic extract of the CTLL and PSCD formulations with an SI value > 10 suppressed *P. falciparum* infection without cytotoxicity. The anti-plasmodial activity and toxicity of both formulations were further investigated in an animal model to confirm the *in vitro* anti-plasmodial results. For the suppression of parasites by the immune system and drug safety before proceeding to clinical trials, *in vivo* models are usually used to test the effect of prodrugs [58]. Although primate models provide a better prediction of the efficacy of antimalarial agents in humans, a mouse model was used as the first step in screening several compounds [59]. Mouse models have been used to identify several conventional antimalarial drugs, including chloroquine, halophantrin, mefloquine, and artemisinin derivatives. Moreover, conducting preliminary pharmacological screening studies has been shown to be more cost-effective than that of the primate model. To evaluate the antimalarial effect, the ICR strain, a common model for the induction of malaria in mice, were inoculated with the wild-type *P. berghei* ANKA strain in this study. The *P. berghei* ANKA strain is a more appropriate parasite with higher penetration and the ability to sequester within blood microcirculation [60]. In this study, the antimalarial activity of the ethanolic extracts of the CTLL and PSCD formulations was

investigated in three models: 4-day suppressive, curative, and prophylactic tests. In all three malaria models, blood parasite suppression was considered the most reliable parameter [61]. Moreover, the plant extract is considered an active compound (possesses an antimalarial effect) when it presents a parasitemia suppression greater than or equal to 30% in a standard screening study [62].

To determine the antimalarial efficacy of compounds, a 4-day suppressive test is commonly used for *in vivo* antimalarial activity screening. This test is widely used to evaluate the efficacy of a compound by comparing blood parasitemia levels between untreated and treated mice [63]. Moreover, the use of *P. berghei*, a rodent malaria protozoan, provides a better tool for predicting drug efficacy for human use in many studies on antimalarial drugs [42]. In the 4-day suppressive test, the ethanolic extracts of the CTLL and PSCD formulations were found to effectively reduce parasitemia levels in a dose-dependent manner. Although the formulation had a chemosuppressive effect, all doses of the extract significantly suppressed parasitemia in mice compared with that in the infected group. However, the percentage suppression of parasitemia of the crude extracts showed a significant difference when compared with artesunate and chloroquine in a 4-day suppressive test. The parasitemia suppression of the extracts demonstrated in the 4-day suppressive test is in agreement with a previous report on ethanolic seed extract of *S. pinnata* (66.82%) [64], methanolic fruit extract of *L. siceraria* (77.37%) [65], ethanolic leaf extract of *B. Pilosa* (74.73%), and aerial extract of *P. lanceolata* (71.29%) [50]. The extracts were identified as active compounds because all treatment showed greater than 30% suppression of parasite levels in the blood. Therefore, these extracts can be indicated as active in schizonticide activity against *P. berghei* ANKA-infected mice. The relative variation in suppressive activity might be due to variations in the content of secondary metabolites, such as polyphenols, flavonoids, alkaloids, terpenoids, and saponins. Thus, the antimalarial effects of the ethanolic extracts could be explained by the possible single or synergistic effects of various active compounds in the extracts [59]. The active compounds related to the antimalaria effect may be associated with antioxidant activity, parasitic DNA damage, immunomodulation, inhibition of protein synthesis, inhibition of parasitic invasion into erythrocytes, prevention of hemozoin formation, and other mechanisms [66].

Since the ethanolic extracts showed significant suppressive activity in a 4-day suppressive test, further investigations were performed to determine their ability to establish parasitic infection using a curative test. All doses of the ethanolic extracts of the CTLL and PSCD formulations at a dose of 200, 400, and 600 mg/kg showed a significant curative effect compared with that in the infected group ($P < 0.05$) with percentages suppression of 15.45%, 31.96%, and 35.94% for the ethanolic extracts of the CTLL formulation and 15.52%, 19.79%, and 29.65% for the ethanolic extracts of the PSCD formulation, respectively. The ethanolic extracts of the CTLL formulation showed significant suppression, with a maximum of 35.94% at 600 mg/kg, indicating a curative effect. A higher dose of the extract had a suppressive effect across the treatment intervals, which exhibited a dose-dependent curative effect on the pathogenesis of malaria. However, the percentage suppression of parasitemia of the crude extracts was significantly reduced compared with that of artesunate and chloroquine in the curative test. Although the administration of the ethanolic extracts of the CTLL and PSCD formulations failed to eradicate the parasite completely, they caused a progressive reduction in parasitemia levels from the first day after treatment to the end of treatment. These results are consistent with those of a previous study on leaf extracts of *Balanaties rotundifolia* (Van Tiegh.) Blattter (Balanitaceae) [67] and *Cordia Africana* (Lam.) (Boraginaceae) in *P. berghei*-infected mice [68]. The delay in antimalarial activity at low doses of crude extracts might be because the extracts administered at low doses did not accumulate sufficiently to induce significant chemosuppression [69].

The curative properties of the ethanolic extracts of the CTLL and PSCD formulations were established in this study. Further investigation was performed to evaluate the prophylactic effect of several traditional plant extracts that exhibited antiplatelet effects in 4-day suppressive and curative tests, as well as prophylactic activity against *P. berghei*-infected mice [70]. In this study, the ethanolic extracts of the CTLL and PSCD formulations showed significant chemo-prophylactic effects against *P. berghei* infection at 600 mg/kg of the CTLL formulation and 400 and 600 mg/kg of the ethanolic extracts of the PSCD formulation compared with that of the untreated control group, with a maximum parasitemia suppression of 65.82% at 600 mg/kg of the ethanolic extracts of the PSCD formulation. The 600 mg/kg ofthe ethanolic extract of the CTLL formulation and 400 and 600 mg/kg of the ethanolic extracts of the PSCD formulation presented significant prophylactic suppression compared with that of the 200 mg/kg of the treated group. Moreover, the percentage parasitemia suppression of the ethanolic extracts of the PSCD formulation at a dose of 600 mg/kg body weight was not significantly different from that of the chloroquine standard drug. The parasitemia suppression of the extracts demon-strated in the prophylactic study is in agreement with a previous report on the leaf extracts of *Calpurnia auraea* (Fabaceae) [71], *Justicia schimperiana* Hochst. EX Nees (Acanthaceae) [72], *Syzygium guineense* (Myrtaceae) [73], and *Cosdia africana* (Lam.) (Boraginaceae) [68]. The prophylactic chemosuppressive level was lower than that of the 4-day suppressive and curative tests. This might be related to the rapid hepatic elimination or active compound component metabolism responsible for the antimalarial activity due to the early administration of the extracts 4 days before infection with *P. berghei* parasite. In the three models of this study, the mice tested throughout all experiments showed no mortality. In addition, the chloroquine standard drug inhibited *P. berghei* parasite growth more ($P < 0.05$) than that of the extract-treated and infected control groups. The lower efficiency of crude extracts might be related to impurities, poor separation, low selectivity, slow-acting absorption, low bioavailability, or other pharmacokinetic and pharmacodynamic parameters [74].

Furthermore, qualitative analysis by GC-MS of the ethanolic extracts of the CTLL and PSCD formulation presented several compounds. The ethanolic extracts of the CTLL formula-tion extract show the presence of different compounds, such as linderol, isoborneol (terpene), β-eudesmol (sesquiterpene), linoleic acid (fatty acid), oleic acid (fatty acid), n-hexadecanoic acid, octadecanoic acid (fatty acid), (+)-camphor, oleamide (organic compound), and agaros-pirol (sesquiterpene). Moreover, GC-MS analysis of the ethanolic extract of the PSCD formu-lation revealed important sources of secondary compounds, including ethyl 4-methoxycinnamate, 3-hydroxy-2-(4-hydroxy-3-methoxyphenyl)-4H-chromen-4-one, fla-menol, oleic acid amide, linoleic acid, oleic acid, ethyl cinnamate, p-heptylphenol, palmitic acid, and pentadecane. Some constituents analyzed by GC-MS are biologically active com-pounds. Remarkably, β-eudesmol is one of the important components that can target several molecular signaling pathways, including Janus kinases/signal transducer and activator of tran-scription proteins, nuclear factor kappa- light-chain enhancer of activated B cells, extracellular signal-regulated kinase, mitogen-activated protein kinase, and the growth factor-associated cAMP response element-binding protein [75]. β-eudesmol has been shown to possess antican-cer, anti-angiogenic, anti-inflammatory, and anti-allergic activities, as well as activities in the gastrointestinal and central nervous system [76]. n-hexadecanoic acid from *Ipomoea eriocarpa* was found to exhibit antioxidant properties against 2,2-diphenyl-1-picrylhydrazyl and 2,2-azi-nobis (3-ethylbenzothiazoline-6-sulfonic acid), and antibacterial activity against *Staphylococ-cus aureus*, *Bacillus subtilis*, *Escherichia coli*, and *Klebsiella pneumoniae* [77]. Certainly, cinnamic acid derivatives are secondary metabolites found in plants and have received much attention in medicinal research on tuberculosis, cardiovascular disease, and malaria [78, 79]. Linoleic acid has shown diverse medicinal properties and potentially beneficial effects in

diseases such as cancer, insulin resistance, skin permeability, cardiovascular disease, and depression. Moreover, linoleic acid has been reported to possess anti-plasmodial properties. The previous study from Paula et al. found that linoleic acid had significantly shown a promising anti-plasmodial activity to D10 and Dd2 strains of the parasite with an $IC_{50} < 10$ μg/mL. Linolenic and linoleic acids suppressed parasitemia levels of 70% and 64%, respectively, against *P. berghei* in a mouse model [80]. Furthermore, Ranjith et al. identified oleamide as a *P. falciparum* thioredoxin reductase ligand from *Guatteria recurvisepala* that exhibited *in vitro* antimalarial activity against *P. falciparum* strain K1 ($IC_{50} = 4.29$ μg/mL) [81]. Therefore, the ethanolic extract of the CTLL and PSCD formulations might be responsible for the antimalarial activity, caused by a single phytoconstituent or the synergistic effect of these compounds, as mentioned above. However, further evaluation is required to isolate, identify, and characterize the active compounds responsible for the observed antimalarial effects.

To test the safety of the extracts, mice were assessed using an oral acute toxicity test. In these toxicity tests, mice were administered a single dose of 2,000 mg/kg of the ethanolic extract of the CTLL and PSCD formulations, where a single high dose is recommended for acute toxicity testing [43]. Behavior, body weight change, biochemical liver and kidney function levels, and histological evaluation were investigated to determine the safety profile of the crude extracts. The acute toxicity results for both ethanolic extracts indicated no signs of toxicity, behavioral changes, or mortality during the 14 days experiment. Therefore, the acute oral $LD_{50}$ of the extracts might be higher than 2,000 mg/kg, identifying its safety effect for antimalarials used according to the OECD guideline No. 425. The results observed in the acute toxicity study with the ethanolic extracts of the CTLL and PSCD formulations were in agreement with those observed in a previous study. These crude extracts at various concentrations of 400–5000 mg/kg body weight indicated safety profiles that showed no any adverse signs of toxicity or mortality in mice [53, 55]. The change in body weight is a SI for toxicity identification after exposure to toxic compounds. In the acute toxicity test results, none of the treatments with the ethanolic extract of CTLL and PSCD formulations showed a significant difference ($P < 0.05$) in body weight change compared with the control group or 7% tween 80 treated groups at the end of this study. This observation indicates that crude extracts do not disturb metabolism in animals. Moreover, the functions of the liver and kidneys have been examined by biochemical analyses to assess the toxicological effects of xenobiotics [82, 83]. Liver and kidney functions were investigated using standard biochemical analyses. AST, ALT, and ALP levels were measured to identify abnormalities in liver function [84]. The plasma ALT levels in mice treated with the ethanolic extracts of the CTLL and PSCD formulations were not significantly different from those in the untreated control and 7% Tween 80 groups. On the other hand, ALP levels were significantly increased in mice treated with the ethanolic extract of the CTLL formulation compared with those in the untreated control and 7% Tween 80 groups. Increased plasma ALP levels may result from hepatic tissue leakage under hepatotoxic conditions [85]. Therefore, the appropriate dose of the ethanolic extract of the CTLL formulation for antimalarial activity needs to be considered. Moreover, this study found an increase in AST levels in mice administered 7% Tween 80 compared with the control group. Elevated AST levels are indicative of cellular necrosis [86]. Therefore, finding a suitable, non-toxic solvent to dissolve the extracts is necessary. BUN and creatinine levels were measured to determine kidney function. BUN and creatinine are both secreted by the kidneys. Excessive plasma BUN and creatinine levels are indicative of kidney cell damage or dysfunction. A reduction in blood supply to the kidney or obstruction of the urinary tract may result in abnormalities in the excretion of BUN and creatinine [87]. These results show that the BUN and creatinine levels in the ethanolic extracts of the CTLL and PSCD formulations were within the normal range and did not differ from those in the untreated control group. Finally, histopathological

examination of the kidney and liver tissues after treatment with the ethanolic extracts of the CTLL and PSCD formulations revealed normal morphological features compared with those in normal healthy mice. Thus, the results suggest that oral administration of crude extracts is neither harmful nor safe.

## Conclusions

The ethanolic extracts of the CTLL and PSCD formulations exhibited high *in vitro* anti-plasmodial activity. In the *in vivo* study, the ethanolic extract of the CTLL formulation (600 mg/kg) exhibited the highest antimalarial activity in the 4-day suppressive and curative tests, whereas the ethanolic extract of the PSCD formulation (600 mg/kg) exhibited the highest antimalarial activity in the prophylactic test. Moreover, a single oral dose of 2,000 mg/kg of the extract showed a safety profile in an acute toxicity model. The anti-plasmodial findings of this study can be used as basic information for scientific research aimed at discovering and developing novel agents with antimalarial activity. Additionally, further studies are required to isolate and identify the active compounds to understand the mechanism of action and evaluate the chronic toxicity of the studied plant.

## Acknowledgments

The authors thank the laboratory workers at the Animal Experiment Building, Walailak University, Thailand, for facilitating the experiments performed on mice. We are grateful to the staff members at the Department of Tropical Pathology, Faculty of Tropical Medicine, Mahidol University, Thailand, for their help with histological processing.

## Author Contributions

**Conceptualization:** Prapaporn Chaniad, Chuchard Punsawad.

**Data curation:** Arisara Phuwajaroanpong, Walaiporn Plirat, Chuchard Punsawad.

**Formal analysis:** Prapaporn Chaniad, Arisara Phuwajaroanpong, Walaiporn Plirat, Atthaphon Konyanee, Chuchard Punsawad.

**Funding acquisition:** Prapaporn Chaniad, Chuchard Punsawad.

**Investigation:** Prapaporn Chaniad, Chuchard Punsawad.

**Methodology:** Prapaporn Chaniad, Arisara Phuwajaroanpong, Walaiporn Plirat, Atthaphon Konyanee, Chuchard Punsawad.

**Project administration:** Prapaporn Chaniad, Chuchard Punsawad.

**Resources:** Prapaporn Chaniad, Chuchard Punsawad.

**Validation:** Prapaporn Chaniad, Chuchard Punsawad.

**Writing – original draft:** Prapaporn Chaniad, Arisara Phuwajaroanpong, Walaiporn Plirat, Chuchard Punsawad.

**Writing – review & editing:** Prapaporn Chaniad, Abdi Wira Septama, Chuchard Punsawad.

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
