## [Decision Letter · Decision Letter 0]

9 Nov 2023

PONE-D-23-19212Assessment of antimalarial activity of crude extract of Chan-Ta-Lee-La and Pra-Sa-Chan-Dang formulations and their plant ingredients for new drug candidates of malaria treatment: in vitro and in vivo experimentsPLOS ONE

Dear Dr. Punsawad,

Thank you for submitting your manuscript to PLOS ONE. After careful consideration, we feel that it has merit but does not fully meet PLOS ONE’s publication criteria as it currently stands. Therefore, we invite you to submit a revised version of the manuscript that addresses the points raised during the review process.

Histopathological photographs should be made in a higher magnification, as suggested by Reviewer #2. Several clarifications in Material and Methods, required by both reviewers, are needed as well. Conclusion seems to fail to be sustained by results provided in Table 9. Please provide species authorities for the species listed in L100-102 as well as for *Artemisia vulgaris* in Table 1. In L323, it is Rane's test instead of Rane test.

We look forward to receiving your revised manuscript.

Kind regards,

Branislav T. Šiler, Ph.D.

Academic Editor

PLOS ONE

Journal Requirements:

 Whilst you may use any professional scientific editing service of your choice, PLOS has partnered with both American Journal Experts (AJE) and Editage to provide discounted services to PLOS authors. Both organizations have experience helping authors meet PLOS guidelines and can provide language editing, translation, manuscript formatting, and figure formatting to ensure your manuscript meets our submission guidelines. To take advantage of our partnership with AJE, visit the AJE website (http://aje.com/go/plos) for a 15% discount off AJE services. To take advantage of our partnership with Editage, visit the Editage website (www.editage.com) and enter referral code PLOSEDIT for a 15% discount off Editage services. If the PLOS editorial team finds any language issues in text that either AJE or Editage has edited, the service provider will re-edit the text for free.

Reviewers' comments:

Reviewer's Responses to Questions

**Comments to the Author**

1. Is the manuscript technically sound, and do the data support the conclusions?

Reviewer #1: Partly

Reviewer #2: Yes

2. Has the statistical analysis been performed appropriately and rigorously? 

Reviewer #1: Yes

Reviewer #2: Yes

3. Have the authors made all data underlying the findings in their manuscript fully available?

Reviewer #1: Yes

Reviewer #2: Yes

4. Is the manuscript presented in an intelligible fashion and written in standard English?

Reviewer #1: Yes

Reviewer #2: Yes

5. Review Comments to the Author

Reviewer #1: General comment:

This study assessed the antimalarial activity of crude extract of Chan-Ta-Lee-La and Pra-Sa-Chan-Dang formulations and their plant ingredients for new drug candidates of malaria treatment both in vitro and in vivo. The paper is well written and the results well presented. The need for scientific appraisal of antimalarial formulations cannot be overemphasized, hence the study is very relevant.

Specific comments to be addressed:

The author must however revise line 274 "eliminating waste from the cages daily" is not the standard practice. Authors must also cite the international guideline(s) being referred to in line 276.

It is also worthy to note that though the mean parasitaemia level was lower than 90% (which is an indication that both formulations were active as antimalarials) in the invivo study, with the % suppression of both CTLL and PSCD being less than 50 (Table 9) the formulations cannot be said to have showed a good antimalarial (curative) activity, hence conclusion is not well justified. Authors must revise the conclusion.

Reviewer #2: • The authors mentioned that “Each plant and formulation 165 were extracted using ethanol and distilled water.”. What was the percentage of the used ethanol?

• Please mention the exact temperature where room temperature varies from country to country “at room temperature for 72 h…..”.

• Please mention the source of Plasmodium falciparum.

• For parasitemia determination “The percentage of parasitemia was determined daily using a light microscope”: I suggest adding a reference (cite) and calculate the percentage of suppression from The antiplasmodial and spleen protective role of crude Indigofera oblongifolia leaf extract traditionally used in the treatment of malaria in Saudi Arabia. Drug Des Devel Ther. 2015 Nov 25;9:6235-46. doi: 10.2147/DDDT.S94673.

• If the used mice were adult (6-8 weeks) add adult before mice.

• Mention how you have calculated 1× 107 P. berghei-parasitized RBCs to be the used dose. Mention the used chamber.

• Did you use the whole blood or you separated the parasitized erythrocytes to be only injected into experimental mice?

• In table 10: “Table 10. Percent suppression of the ethanolic extract of the CTLL and PSCD formulations in prophylactic test” mention the day of blood collection. Why this day.

• In the histopathology: I can not see any histopathological change with this magnification. I found no figure caption describing the groups and the histopathological change. I can not see the scale bar. I also suggest in addition to add a higher magnification to add a histology score.

• I suggest comparing your results with authors used any of the used extracts or mention the activity of other extracts and write on the importants of using medithinal plants against malaria. You can use “ Medicinal plants as a fight against murine blood-stage malaria. Saudi J Biol Sci. 2021 Mar;28(3):1723-1738. doi: 10.1016/j.sjbs.2020.12.014.

6. PLOS authors have the option to publish the peer review history of their article (what does this mean?). If published, this will include your full peer review and any attached files.

Reviewer #1: **Yes: **Samuel Adjei

Reviewer #2: No

---

## [Author Response · Author response to Decision Letter 0]

25 Nov 2023

POINT-BY-POINT RESPONSES TO THE REVIEWERS’ COMMENTS

Editor

Histopathological photographs should be made in a higher magnification, as suggested by Reviewer #2. Several clarifications in Material and Methods, required by both reviewers, are needed as well. Conclusion seems to fail to be sustained by results provided in Table 9. Please provide species authorities for the species listed in L100-102 as well as for Artemisia vulgaris in Table 1. In L323, it is Rane's test instead of Rane test.

Response: The manuscript has been revised as suggested. 

Reviewer #1: 

General comment: This study assessed the antimalarial activity of crude extract of Chan-Ta-Lee-La and Pra-Sa-Chan-Dang formulations and their plant ingredients for new drug candidates of malaria treatment both in vitro and in vivo. The paper is well written and the results well presented. The need for scientific appraisal of antimalarial formulations cannot be overemphasized, hence the study is very relevant.

Response: Thank you.

Specific comments to be addressed:

The author must however revise line 274 "eliminating waste from the cages daily" is not the standard practice. Authors must also cite the international guideline(s) being referred to in line 276.

Response: The sentence has been revised. Please see page 13, line 275.

It is also worthy to note that though the mean parasitaemia level was lower than 90% (which is an indication that both formulations were active as antimalarials) in the in vivo study, with the % suppression of both CTLL and PSCD being less than 50 (Table 9) the formulations cannot be said to have showed a good antimalarial (curative) activity, hence conclusion is not well justified. Authors must revise the conclusion.

Response: The conclusion has been revised. Please see page 43, line 857-859 and page 3, lines 49-51.

Reviewer #2: 

• The authors mentioned that “Each plant and formulation 165 were extracted using ethanol and distilled water.”. What was the percentage of the used ethanol?

Response: The percentage of ethanol at 95% has been added to the manuscript. Please see page 8, line 165.

• Please mention the exact temperature where room temperature varies from country to country “at room temperature for 72 h…..”.

Response: Room temperature at 25 °C has been added to the manuscript. Please see page 9, line 169.

• Please mention the source of Plasmodium falciparum.

Response: The sentence about the source of Plasmodium falciparum has been added to the manuscript. Please see page 9, lines 182-184.

• For parasitemia determination “The percentage of parasitemia was determined daily using a light microscope”: I suggest adding a reference (cite) and calculate the percentage of suppression from The antiplasmodial and spleen protective role of crude Indigofera oblongifolia leaf extract traditionally used in the treatment of malaria in Saudi Arabia. Drug Des Devel Ther. 2015 Nov 25;9:6235-46. doi: 10.2147/DDDT.S94673.

Response: A suggested reference (cite) has been added. Please see page 9, line 190.

• If the used mice were adult (6-8 weeks) add adult before mice.

Response: The word “adult” has been added to the manuscript. Please see page 13, line 268.

• Mention how you have calculated 1× 107 P. berghei-parasitized RBCs to be the used dose. Mention the used chamber.

Response: For the calculation of 1× 107 P. berghei-parasitized RBCs to be the used dose, we calculated from the level of parasitemia in donor mice. When the level of parasitemia in donor mice reached 20–30%, the donor mouse was subsequently sacrificed and blood collected (via cardiac puncture) into a heparinized tube. Based on the parasitemia level of the donor mouse and the red blood cell (RBC) count of a normal mouse, the collected blood was diluted with normal saline (0.9%), so that 1 ml of blood contains 5 x 107 infected RBCs. Each mouse was injected with 0.2 ml of blood containing 1 x 107 P. berghei-infected RBCs via the intraperitoneal route.

Ref: Belay, W. Y., Endale Gurmu, A., & Wubneh, Z. B. Antimalarial activity of stem bark of Periploca linearifolia during early and established Plasmodium infection in mice. J Evid Based Complementary Altern Med.2018; 4169397. https://doi.org/10.1155/2018/4169397

• Did you use the whole blood or you separated the parasitized erythrocytes to be only injected into experimental mice?

Response: For the infection of mice, we used whole blood. Briefly, blood from donor mice was collected and then diluted with 0.9% physiological isotonic saline before injected into mice in the experiment.

• In table 10: “Table 10. Percent suppression of the ethanolic extract of the CTLL and PSCD formulations in prophylactic test” mention the day of blood collection. Why this day.

Response: 

• The word “Day 6” has been added to the table name. Please see page 30, line 563. 

• We monitored parasitemia on day 6 (day 6 after mice received the extract or day 3 post-infection) because the parasitemia level increased steadily beginning on day 3 post-infection. The reason could be that only approximately one-third of infected red blood cells are able to circulate freely in the blood circulation within 48 hours, whereas the remaining parasites stick to the venules and capillaries. Therefore, blood collection at day 3 post-infection would provide a precise assessment of the parasitemia. 

Ref: Nwonuma CO, Balogun EA, Gyebi GA. Evaluation of antimalarial activity of ethanolic extract of Annona muricata L.: An in vivo and an in-silico approach. Journal of Evidence-Based Integrative Medicine. 2023;28. 

• In the histopathology: I cannot see any histopathological change with this magnification. I found no figure caption describing the groups and the histopathological change. I cannot see the scale bar. I also suggest in addition to add a higher magnification to add a histology score.

Response: Thank you for your suggestion. A new figure of histopathological change with a higher magnification has been added.

• I suggest comparing your results with authors used any of the used extracts or mention the activity of other extracts and write on the important of using medithinal plants against malaria. You can use “Medicinal plants as a fight against murine blood-stage malaria. Saudi J Biol Sci. 2021 Mar;28(3):1723-1738. doi: 10.1016/j.sjbs.2020.12.014.

Response: Thank you for your valuable comment. I have already compared results with authors who used any of the used extracts or mentioned the activity of other extracts (please see pages 37–38, lines 720–723) and wrote about the importance of using medicinal plants against malaria (please see pages 34, lines 63–63).

---

## [Editor Report · Decision Letter 1]

7 Dec 2023

PONE-D-23-19212R1Assessment of antimalarial activity of crude extract of Chan-Ta-Lee-La and Pra-Sa-Chan-Dang formulations and their plant ingredients for new drug candidates of malaria treatment: in vitro and in vivo experimentsPLOS ONE

Dear Dr. Punsawad,

Thank you for submitting your manuscript to PLOS ONE. After careful consideration, we feel that it has merit but does not fully meet PLOS ONE’s publication criteria as it currently stands. Therefore, we invite you to submit a revised version of the manuscript that addresses the points raised during the review process.

**Authors have overseen the Editor's request provided in the previous review round "**Please provide species authorities for the species listed in L100-102 as well as for *Artemisia vulgaris* in Table 1.**" You may use Kew Garden's Plants of the World Online (**https://powo.science.kew.org/**).**

We look forward to receiving your revised manuscript.

Kind regards,

Branislav T. Šiler, Ph.D.

Academic Editor

PLOS ONE
---

## [Author Response · Author response to Decision Letter 1]

10 Dec 2023

The Editor's request:

"Please provide species authorities for the species listed in L100-102 as well as for Artemisia vulgaris in Table 1." You may use Kew Garden's Plants of the World Online (https://powo.science.kew.org/).

Response: We have checked and revised as suggested.

Response: We have checked and revised as suggested.

---

## [Editor Report · Decision Letter 2]

18 Dec 2023

Assessment of antimalarial activity of crude extract of Chan-Ta-Lee-La and Pra-Sa-Chan-Dang formulations and their plant ingredients for new drug candidates of malaria treatment: in vitro and in vivo experiments

PONE-D-23-19212R2

Dear Dr. Punsawad,

We’re pleased to inform you that your manuscript has been judged scientifically suitable for publication and will be formally accepted for publication once it meets all outstanding technical requirements.

Kind regards,

Branislav T. Šiler, Ph.D.

Academic Editor

PLOS ONE
---

## [Editor Report · Acceptance letter]

3 Jan 2024

PONE-D-23-19212R2 

PLOS ONE

Dear Dr. Punsawad, 

I'm pleased to inform you that your manuscript has been deemed suitable for publication in PLOS ONE. Congratulations! Your manuscript is now being handed over to our production team.

Kind regards, 

on behalf of

Dr. Branislav T. Šiler 

Academic Editor

PLOS ONE